# GENERATIVE FLOWS ON SYNTHETIC PATHWAY FOR DRUG DESIGN

**Seonghwan Seo**[1*], **Minsu Kim**[1], **Tony Shen**[2], **Martin Ester**[2], **Jinkyoo Park**[1,3],
**Sungsoo Ahn**[4], **Woo Youn Kim**[1,5*]
[1]KAIST  [2]Simon Fraser University  [3]OMELET  [4]POSTECH  [5]HITS

## ABSTRACT

Generative models in drug discovery have recently gained attention as efficient alternatives to brute-force virtual screening. However, most existing models do not account for synthesizability, limiting their practical use in real-world scenarios. In this paper, we propose RXNFLOW, which sequentially assembles molecules using predefined molecular building blocks and chemical reaction templates to constrain the synthetic chemical pathway. We then train on this sequential generating process with the objective of generative flow networks (GFlowNets) to generate both highly rewarded and diverse molecules. To mitigate the large action space of synthetic pathways in GFlowNets, we implement a novel action space subsampling method. This enables RXNFLOW to learn generative flows over extensive action spaces comprising combinations of 1.2 million building blocks and 71 reaction templates without significant computational overhead. Additionally, RXNFLOW can employ modified or expanded action spaces for generation without retraining, allowing for the introduction of additional objectives or the incorporation of newly discovered building blocks. We experimentally demonstrate that RXNFLOW outperforms existing reaction-based and fragment-based models in pocket-specific optimization across various target pockets. Furthermore, RXNFLOW achieves state-of-the-art performance on CrossDocked2020 for pocket-conditional generation, with an average Vina score of –8.85 kcal/mol and 34.8% synthesizability.

## 1 INTRODUCTION

Structure-based drug discovery (SBDD) has emerged as a pivotal paradigm for early drug discovery, facilitated by the increasing accessibility of protein structure prediction tools (Jumper et al., 2021) and high-resolution crystallography (Liu et al., 2015). However, traditional brute-force virtual screening is computationally expensive (Graff et al., 2021), prompting the development of deep generative models that can bypass this inefficiency. In this context, various approaches such as deep reinforcement learning (Zhavoronkov et al., 2019), variational autoencoders (Zhung et al., 2024) generative adversarial network (Ragoza et al., 2022), and diffusion models (Guan et al., 2023a;b) have been proposed to directly sample candidate molecules against a given protein structure.

While generative models have shown success in molecular discovery with desirable biological properties, most overlook synthesizability which is a crucial factor for wet-lab validation (Gao & Coley, 2020). One line to improve synthesizability is multi-objective optimization using cheap functions to estimate the synthesizability (Ertl & Schuffenhauer, 2009), but this is too simplified to reflect complex synthetic principles (Cretu et al., 2024). Other efforts aim to project molecules from generative models into a synthesizable space (Gao et al., 2022b; Luo et al., 2024; Gao et al., 2024b), but chemical modifications in this process can degrade the optimized properties.

To address this issue, recent works formulate the generation of synthetic pathways as a Markov decision process (MDP) for molecular design (Gottipati et al., 2020). These approaches return a synthesizable molecule by assembling purchasable building blocks and reaction templates according to the generated synthetic pathway. Notably, the emergence of virtual libraries—created by combinatorially enumerating building blocks and reaction templates, such as Enamine REAL (Grygorenko et al., 2020)—allows the generated molecules to be readily synthesizable on demand with

---

*Correspondence to {`shwan0106`, `wooyoun`}@kaist.ac.kr

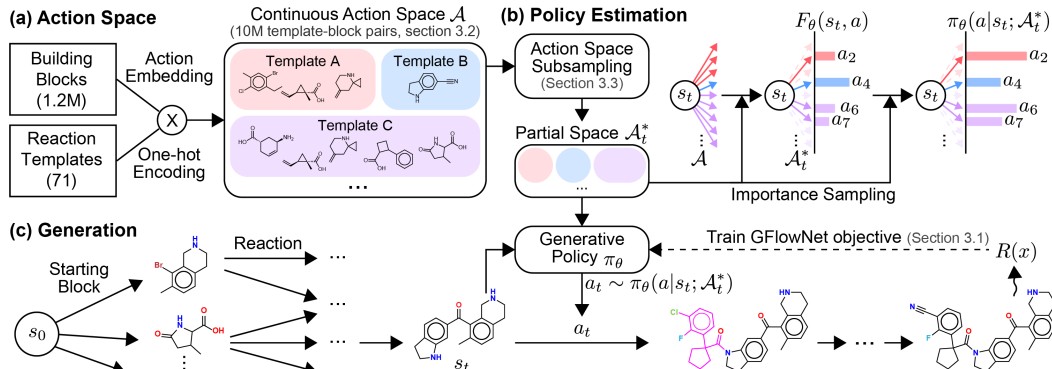

Figure 1: **Overview of RXNFLOW. (a)** Synthetic action space which is represented in a continuous action space. Each colored box corresponds to a reaction template and the molecules in the box are reactant blocks. **(b)** Policy estimation using the action space subsampling in a manner of importance sampling. **(c)** Molecular generation process and model training.

their synthetic pathways. Recent studies (Cretu et al., 2024; Koziarski et al., 2024) advanced this approach by training the decision-making policy using the objective of generative flow networks (GFlowNets; Bengio et al., 2021). This objective encourages the policy to sample in proportion to the reward function, enabling the retrieval of samples from a diverse range of modes.

Unlike atom-based or fragment-based models, the synthetic action spaces are massive and composed of millions of building blocks and tens of reaction templates. While the large action spaces offer opportunities to discover novel hit candidates by expanding an explorable chemical space (Sady-bekov et al., 2022), it incurs significant computational overhead. Thus, prior works have restricted the action spaces to trade off the size of the search space for efficiency. However, reducing the search space leads to a decrease in diversity and synthetic complexity. While one could compensate for the reduced number of building block candidates by adding more reaction steps, this leads to the increase in synthetic complexity, negatively influencing synthesizability, yield, and cost (Coley et al., 2018; Kim et al., 2023).

In response to this challenge, we propose RXNFLOW, a synthesis-oriented generative framework that allows training generative flows over a large action space to generate synthetic pathways for drug design. The distinctive features of this method are as follows. First, we introduce an *action space subsampling* (Figure 1) to handle massive action spaces without significant memory overhead, enabling us to explore a broader chemical space with fewer reaction steps than existing models. Then, we train the generative policy with a GFlowNet objective to sample both diverse and potent molecules from the expanded search space. We demonstrate that RXNFLOW effectively generates drug candidates, outperforming existing reaction-based, atom-based, and fragment-based baselines across various SBDD tasks, while ensuring the synthesizability of generated drug candidates. We also achieve a new state-of-the-art Vina score, drug-likeness, and synthesizability on the Cross-Docked2020 pocket-conditional generation benchmark (Luo et al., 2021).

Furthermore, we formulate an adaptable MDP (Figure 2) for consistent flow estimation on modified building block libraries, which can be highly practical in real-world applications. By combining the proposed MDP with action embedding (Dulac-Arnold et al., 2015), which represents actions in a continuous space instead of a discrete space, RXNFLOW can achieve further objectives or incorporate newly discovered building blocks without retraining. We experimentally show that RXNFLOW can achieve an additional solubility objective and behave appropriately for unseen building blocks. This capability makes RXNFLOW highly adaptable to real-world drug discovery pipeline, where new objectives frequently arise (Fink et al., 2022) and building block libraries are continuously expanding (Grygorenko et al., 2020).

## 2 RELATED WORKS

**Structure-based drug discovery.** The first type of SBDD involves **pocket-specific optimization** methods to enhance docking scores against a single pocket, including evolutionary algorithms (Rei-

denbach, 2024), reinforcement learning (RL) (Zhavoronkov et al., 2019), and GFlowNets (Bengio et al., 2021). However, these require individual optimizations for each pocket, limiting scalability. The second type is based on **pocket-conditional generation**, which generates molecules against arbitrary given pockets without additional training. This can be achieved by distribution-based generative models (Ragoza et al., 2022; Peng et al., 2022; Guan et al., 2023b; Schneuing et al., 2023; Qu et al., 2024) trained on protein-ligand complex datasets to model ligand distributions for given pockets. On the other hand, Shen et al. (2023) formulated a pocket-conditioned policy for GFlowNets to generate samples from reward-biased distributions in a *zero-shot* manner.

**Syntheis-oriented generative models.** To ensure the synthesizability of generated molecules, synthesis-oriented *de novo* design approaches incorporate combinatorial chemistry principles into generative models. Bradshaw et al. (2019) represented the synthetic pathways as directed acyclic graphs (DAG) for generative modeling. Horwood & Noutahi (2020) formulated the synthetic pathway generation as an MDP and optimized the molecules with RL. Similarly, Gao et al. (2022b) employed a genetic algorithm to optimize synthesis trees to generate molecules with the desired properties. Seo et al. (2023) proposed a conditional generative model to directly sample molecules with desired properties without optimization. Recently, Cretu et al. (2024); Koziarski et al. (2024) proposed the reaction-based GFlowNet to generate diverse and potent molecules.

**Action embedding for large action spaces.** To handle large action spaces in the synthesis-oriented generation, Gottipati et al. (2020) employed action embedding (Dulac-Arnold et al., 2015) that represents building blocks in a continuous action space with their chemical information. Later, Seo et al. (2023); Koziarski et al. (2024) experimentally demonstrated that it can enhance the model training and generative performance. The continuous action space provides the benefit of reducing the computational complexity for sampling from large space of actions and the memory requirement for parameterizing the categorical distribution over the large action space.

**Generative Flow Networks.** GFlowNets are a learning framework for a stochastic generative policy that constructs an object through a series of decisions, where the probability of generating each object is proportional to a given reward associated with that object (Bengio et al., 2021). Unlike other optimization methods that maximize rewards and often converge to a single solution, GFlowNets aim to sample a diverse set of high-rewarded modes, which is vital for novel drug design (Shen et al., 2023; Jain et al., 2022). To this end, the generative policy is trained using objectives such as flow matching (Bengio et al., 2021), detailed balance (Bengio et al., 2023), and trajectory balance (Malkin et al., 2022). Extending the GFlowNets to various applications is an active area of research, e.g., GFlowNets have been applied to designing crystal structures (Nguyen et al., 2023b), phylogenetic inference (Zhou et al., 2023), finetuning diffusion models (Venkatraman et al., 2024), and causal inference (Nguyen et al., 2023a).

## 3 METHOD

### 3.1 GFLOWNET PRELIMINARIES

GFlowNets (Bengio et al., 2021) are the class of generative models that learn to sample objects $x \in \mathcal{X}$ proportional to a given reward function, i.e., $p(x) \propto R(x)$. This is achieved by sequentially constructing a compositional object $x$ through a series of state transitions $s \rightarrow s'$, forming a trajectory $\tau = (s_0 \rightarrow \ldots \rightarrow s_n = x) \in \mathcal{T}$. The set of all complete trajectories from the initial state $s_0$ can be represented as a directed acyclic graph $\mathcal{G} = (\mathcal{S}, \mathcal{A})$ with a reachable state space $\mathcal{S}$ and an action space $\mathcal{A}$. Each action $a$ induces a transition from the state $s$ to the state $s'$, expressed as $s' = T(s, a)$ and represented as $s \rightarrow s'$. Then, we define the *trajectory flow* $F(\tau)$, which flows along the trajectory $\tau = (s_0 \rightarrow \ldots \rightarrow s_n = x)$, as the reward of the terminal state, $R(x)$. The *edge flow* $F(s \rightarrow s')$, or equivalently $F(s, a)$, is defined as the total flow along the edge $a : s \rightarrow s'$:

$$F(s \rightarrow s') = F(s, a) = \sum_{\tau \in \mathcal{T} \text{ s.t. } (s \rightarrow s') \in \tau} F(\tau). \tag{1}$$

The *state flow* $F(s)$ for the intermediate state is defined as the total flow through the state $s$:

$$F(s) = \sum_{\tau \in \mathcal{T} \text{ s.t. } s \in \tau} F(\tau) = \underbrace{\sum_{(s'' \rightarrow s) \in \mathcal{A}} F(s'' \rightarrow s) = \sum_{(s \rightarrow s') \in \mathcal{A}} F(s \rightarrow s')}_{\text{Intermediate flow matching condition}}. \tag{2}$$

In addition to the flow matching condition for intermediate states, there are two boundary conditions for the states. First, the flow of a terminal state $x$ must equal the reward of the objective: $F(x) = R(x)$. Second, the *partition function*, $Z$, is equivalent to the sum of all trajectory flows and the sum of all rewards: $Z = F(s_0) = \sum_{\tau \in \mathcal{T}} F(\tau) = \sum_{x \in \mathcal{X}} R(x)$. These three conditions—one for intermediate states and two for boundary states—are known as the *flow matching* conditions and ensure that GFlowNets generate objectives proportional to their rewards.

To convert the flow network into a usable policy, we define the *forward policy* as the forward transition probability $P_F(s'|s)$ and the *backward policy* as the backward transition probability $P_B(s|s')$:

$$P_F(s'|s) := P(s \to s'|s) = \frac{F(s \to s')}{F(s)}, \quad P_B(s|s') := P(s \to s'|s') = \frac{F(s \to s')}{F(s')}. \quad (3)$$

## 3.2 ACTION SPACE FOR SYNTHETIC PATHWAY GENERATION

Following Cretu et al. (2024), we treated a chemical reaction as a forward transition and a synthetic pathway as a trajectory for molecular generation. For the initial state $s_0$, the model always chooses `AddFirstReactant` to sample a building block $b$ from the entire building block set $\mathcal{B}$ as a starting molecule. For the later states $s$, the model samples actions among `ReactUni`, `ReactBi`, or `Stop`. When the action type is `ReactUni`, the model performs *in silico* uni-molecular reactions with an assigned reaction template $r \in \mathcal{R}_1$. When the action type is `ReactBi`, the model performs bi-molecular reactions with a reaction template $r \in \mathcal{R}_2$ and a reactant block $b$ in the possible reactant set for the reaction template $r$: $\mathcal{B}_r \subseteq \mathcal{B}$. If `Stop` is sampled, the trajectory is terminated. To sum up, the allowable action space $\mathcal{A}(s)$ for the state $s$ is:

$$\mathcal{A}(s) = \begin{cases} \mathcal{B} & \text{if } s = s_0 \\ \{\texttt{Stop}\} \cup \mathcal{R}_1 \cup \{(r,b)|r \in \mathcal{R}_2, b \in \mathcal{B}_r\} & \text{otherwise} \end{cases} \quad (4)$$

where unavailable reaction templates to the molecule of the state $s$ are masked.

## 3.3 FLOW NETWORK ON ACTION SPACE SUBSAMPLING

We propose a novel memory-efficient technique called the *action space subsampling*, that estimates the state flow $F_\theta(s)$ from a subset of the outgoing edge flows $F_\theta(s \to s')$ for forward policy estimation. First, we implement an auxiliary policy, termed *subsampling policy* $\mathcal{P}(\mathcal{A})$, which samples a subset of the action space $\mathcal{A}^* \subseteq \mathcal{A}$. This reduces both the memory footprint and the computational complexity from $\mathcal{O}(|\mathcal{B}||\mathcal{R}_2|)$ to $\mathcal{O}(|\mathcal{B}^*||\mathcal{R}_2|)$ with the controllable size $|\mathcal{B}^*|$. We then estimate the forward policy by importance sampling. In contrast to the parameterized forward policy, we formulate a fixed backward policy since it is hard to force invariance to molecule isomorphism (Malkin et al., 2022). Theoretical backgrounds are provided in Sec. A.

**Subsampling policy.** Subsampling policy $\mathcal{P}(\mathcal{A})$ performs uniform sampling for the initial state and importance sampling for the later states. For the initial state, the allowable action space $\mathcal{A}(s_0) = \mathcal{B}$ is homogeneous since all of them are `AddFirstReactant` actions. For the later state, the action space is comprised of one `Stop`, tens of `ReactUni` actions, and millions of `ReactBi` actions. To capture rare-type actions in the inhomogeneous space, we use all `Stop` and `ReactUni` actions. The partial action space $\mathcal{A}^*(s) \sim \mathcal{P}(\mathcal{A}(s))$ comprises the uniform subset $\mathcal{B}^* \subseteq \mathcal{B}$ or $\mathcal{B}_r^* \subseteq \mathcal{B}_r$:

$$\mathcal{A}^*(s) = \begin{cases} \mathcal{B}^* & \text{if } s = s_0 \\ \{\texttt{Stop}\} \cup \mathcal{R}_1 \cup \{(r,b)|r \in \mathcal{R}_2, b \in \mathcal{B}_r^*\} & \text{otherwise} \end{cases} \quad (5)$$

**Forward policy.** To estimate the forward policy $P_F(s'|s) = F(s \to s')/F(s)$ from the partial action space $\mathcal{A}^*$, we estimate the state flow $F(s)$ with a subset of outgoing edge flows $F(s \to s')$. Since we introduce importance sampling for action types, we weight the edge flow $F(s \to s')$ of edge $a : s \to s'$ according to the subsampling ratio:

$$w_a = w_{(s \to s')} = \begin{cases} |\mathcal{B}|/|\mathcal{B}^*| & \text{if } a \in \mathcal{B} \\ |\mathcal{B}_r|/|\mathcal{B}_r^*| & \text{if } a \text{ is } (r,b) \text{ where } r \in \mathcal{R}_2 \\ 1 & \text{if } a \text{ is } \texttt{Stop} \text{ or } a \in \mathcal{R}_1 \end{cases} \quad (6)$$

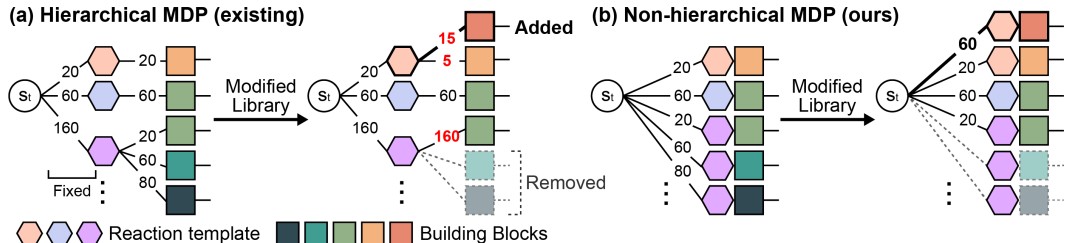

Figure 2: **Comparison of using modified building block library for the generation**: **(a)** a hierarchical MDP, and **(b)** a non-hierarchical MDP. More details are in Figure 8.

By weighting edge flows, we can estimate the state flow $\hat{F}_\theta(s; \mathcal{A}^*)$ as:

$$\hat{F}_\theta(s; \mathcal{A}^*) = \sum_{(s \to s') \in \mathcal{A}^*(s)} w_{(s \to s')} F_\theta(s \to s'), \tag{7}$$

which the estimated forward policy is $\hat{P}_F(s'|s; \mathcal{A}^*; \theta) = F_\theta(s \to s')/\hat{F}_\theta(s; \mathcal{A}^*)$.

**Action embedding.** In the standard implementation (Bengio et al., 2021) of the flow function $F_\theta$ and its neural network $\phi_\theta$, the edge flow of the edge $a : s \to s'$ is computed with the corresponding action-specific parameter $\theta_a$: $F_\theta(s \to s') = F_\theta(s, a) = \phi_{\theta_a}^{\text{flow}}(\phi_\theta^{\text{state}}(s))$.

However, large action spaces require numerous parameters which increase model complexity. To address this, we use an additional network $\phi_\theta^{\text{block}}$ for `AddFirstReactant` and `ReactBi`, which embeds the building block $b$ into a continuous action space with its structural information, molecular fingerprints (see Sec. B.3):

$$F_\theta(s_0, b) = \phi_\theta^{\text{flow}}(\phi_\theta^{\text{state}}(s), \phi_\theta^{\text{block}}(b)), \quad F_\theta(s, (r, b)) = \phi_\theta^{\text{flow}}(\phi_\theta^{\text{state}}(s), \delta(r), \phi_\theta^{\text{block}}(b)) \tag{8}$$

where $\delta(r)$ is the one-hot encoding for a bi-molecular reaction template $r$.

**GFlowNet training.** In this work, we use the trajectory balance (TB; Malkin et al., 2022) as the training objective of GFlowNets from Eq. (9) and train models following Sec. B.4. The action space subsampling is performed for each transition $s_t \to s_{t+1}$: $\mathcal{A}_t^* \sim \mathcal{P}(\mathcal{A})$.

$$\hat{\mathcal{L}}_{\text{TB}}(\tau) = \left( \log \frac{Z_\theta \prod_{t=1}^n \hat{P}_F(s_t|s_{t-1}; \mathcal{A}_{t-1}^*; \theta)}{R(x) \prod_{t=1}^n P_B(s_{t-1}|s_t)} \right)^2 \tag{9}$$

For online training, we use the sampling policy $\pi_\theta$ proportional to $\hat{P}_F(-|-; \mathcal{A}^*; \theta)$, given by:

$$\pi_\theta(s'|s; \mathcal{A}^*) = \frac{w_{(s \to s')} F_\theta(s \to s')}{\sum_{(s \to s'') \in \mathcal{A}^*(s)} w_{(s \to s'')} F_\theta(s \to s'')} \tag{10}$$

### 3.4 JOINT SELECTION OF TEMPLATES AND BLOCKS

For bi-molecular reactions, existing synthesis-oriented methods (Gao et al., 2022b; Cretu et al., 2024; Koziarski et al., 2024) formulated a hierarchical MDP which selects a reaction template $r$ first and then the corresponding reactant block $b$ sequentially from Eq. (11). However, as shown in Figure 2, the probability of selecting each reaction template $r$ is fixed after training in a hierarchical MDP, and this rigidity can lead to incorrect policy estimates in modified block libraries. Therefore, we formulate a non-hierarchical MDP that jointly selects reaction templates and reactant blocks $(r, b)$ at once, as given by Eq. (12), resulting in more consistent estimates of forward policy $P_F$.

$$P_F(T(s, (r, b))|s; \theta) = \frac{F_\theta(s, r)}{\sum_{r' \in \mathcal{R}_1 \cup \mathcal{R}_2 \cup \{\text{Stop}\}} F_\theta(s, r')} \times \frac{F_\theta(s, (r, b))}{\sum_{b' \in \mathcal{B}_r} F_\theta(s, (r, b'))} \tag{11}$$

$$P_F(T(s, (r, b))|s; \theta) = \frac{F_\theta(s, (r, b))}{\sum_{r' \in \mathcal{R}_1 \cup \{\text{Stop}\}} F_\theta(s, r') + \sum_{r' \in \mathcal{R}_2} \sum_{b' \in \mathcal{B}_{r'}} F_\theta(s, (r', b'))} \tag{12}$$

## 4 EXPERIMENTS

**Overview.** We validate the effectiveness of RXNFLOW in two common SBDD tasks: pocket-specific optimization (Sec. 4.1) and pocket-conditional generation (Sec. 4.2). To the best of our knowledge, this is the first synthesis-oriented approach for pocket-conditional generation. We also investigate the applicability of RXNFLOW in real-world drug discovery pipelines where new further objectives may be introduced (Sec. 4.3) and the building block libraries are constantly expanded (Sec. 4.4). Lastly, we conduct an ablation study in Sec. 4.5 and a theoretical analysis in Sec. D.8. Source code available at `https://github.com/SeonghwanSeo/RxnFlow`.

**Setup.** We use the reaction template set constructed by Cretu et al. (2024) including 13 uni- and 58 bi-molecular reaction templates. For the building blocks, we use 1.2M blocks from the Enamine comprehensive catalog. We use up to 3 reaction steps for generation following Enamine REAL Space (Grygorenko et al., 2020), while SynFlowNet and RGFN allow 4 steps. For the subsampling policy, we set a sampling ratio of 1%. The experimental details are provided in Sec. C.

**Synthesizability estimation.** To assess the synthesizability of the generated compounds, we used the computationally intensive retrosynthetic analysis tool AiZynthFinder (Genheden et al., 2020) with the Enamine building block library. We note that the molecule is identified as synthesizable only if it can be synthesized using the USPTO reactions (Lowe, 2017) and given building blocks.

### 4.1 POCKET-SPECIFIC OPTIMIZATION WITH GPU-ACCELERATED DOCKING

**Setup.** Since GFlowNets sample a large number of molecules for online training, we employed a GPU-accelerated UniDock (Yu et al., 2023) with Vina scoring (Trott & Olson, 2010). It is well known that docking can be hacked by increasing molecule size (Pan et al., 2003), so the appropriate constraints are required. We select QED (Bickerton et al., 2012) as a comprehensive molecular property constraint, QED$>$0.5, and set the reward function as $R(x) = w_1 \text{QED}(x) + w_2 \widehat{\text{Vina}}(x)$ where $w_1, w_2$ are used as the input of multi-objective GFlowNets (Jain et al., 2023) for all GFlowNets and are set to 0.5 for non-GFlowNet baselines. $\widehat{\text{Vina}}$ is a normalized docking score (Eq. (32)).

Each method generates up to 64,000 molecules for each of the 15 proteins in the LIT-PCBA dataset (Tran-Nguyen et al., 2020). We then filter the molecules with the property constraint and select the top 100 diverse candidates based on the docking score, using a Tanimoto distance threshold of 0.5 to ensure structural diversity. The selected molecules are evaluated with the following metrics: **Hit ratio (%)** measures the fraction of *hits*, defined as the molecules that are identified as synthesizable by AiZynthFinder and having better docking scores than known reference ligands (Lee et al., 2023). **Vina (kcal/mol)** measures the average docking score. **Synthesizability (%)** is the fraction of synthesizable molecules identified by AiZynthFinder. **Synthetic complexity**, which is highly correlated to yield and cost, is evaluated as the average number of synthesis steps (Coley et al., 2018).

**Baselines.** We perform comparisons to various synthetic-oriented approaches: genetic algorithm (**SynNet**) (Gao et al., 2022b), conditional generative model (**BBAR**) (Seo et al., 2023), and GFlowNets (**SynFlowNet**, **RGFN**) (Cretu et al., 2024; Koziarski et al., 2024). For SynFlowNet and RGFN, we used 6,000 and 350 blocks, respectively, and set the maximum reaction step to 4 following the original papers. Moreover, we consider fragment-based GFlowNets (**FragGFN**) to analyze the effects of synthetic constraints on the performance. For FragGFN, we also consider additional synthesizability objectives with commonly-used synthetic accessibility score (SA; Ertl & Schuffenhauer, 2009) (**FragGFN+SA**).

**Results.** The results for the first five targets are shown in Tables 1 and 2, and additional results for the 10 remaining targets are reported in Sec. D.1. The property distribution for each target are reported in Sec. D.2. RXNFLOW outperforms the baselines across all test proteins, demonstrating that the expanded sample space with the large action space enabled the model to generate more potent and diverse molecules. Additionally, as shown in Tables 3 and 4, RXNFLOW ensures the synthesizability of the generated molecules more effectively than the other synthesis-oriented methods and GFlowNets which employ the same reactions as ours. These results support our primary assertion that existing synthesis-oriented approaches using more synthetic steps with smaller building block libraries can increase overall synthesis complexity and reduce synthesizability. Furthermore, Frag-

Table 1: **Hit ratio.** Mean and standard deviation over 4 runs. The best results are in bold.

| Category | Method | Hit Ratio (%, ↑) | | | | |
| --- | --- | --- | --- | --- | --- | --- |
| | | ADRB2 | ALDH1 | ESR_ago | ESR_antago | FEN1 |
| Fragment | FragGFN | 4.00 (± 3.54) | 3.75 (± 1.92) | 0.25 (± 0.43) | 0.25 (± 0.43) | 0.25 (± 0.43) |
| | FragGFN+SA | 5.75 (± 1.48) | 4.00 (± 1.58) | 0.25 (± 0.43) | 0.00 (± 0.00) | 0.00 (± 0.00) |
| Reaction | SynNet | 45.83 (± 7.22) | 25.00 (± 25.00) | 0.00 (± 0.00) | 0.00 (± 0.00) | 50.00 (± 0.00) |
| | BBAR | 21.25 (± 5.36) | 18.25 (± 1.92) | 3.50 (± 1.12) | 2.25 (± 1.09) | 11.75 (± 2.59) |
| | SynFlowNet | 52.75 (± 1.09) | 57.00 (± 6.04) | 30.75 (± 10.03) | 11.25 (± 1.48) | 53.00 (± 8.92) |
| | RGFN | 46.75 (± 6.87) | 39.75 (± 8.17) | 4.50 (± 1.66) | 1.25 (± 0.43) | 19.75 (± 4.32) |
| | **RXNFLOW** | **60.25** (± 3.77) | **63.25** (± 3.11) | **71.25** (± 4.15) | **46.00** (± 7.00) | **65.50** (± 4.09) |

Table 2: **Vina.** Mean and standard deviation over 4 runs. The best results are in bold.

| Category | Method | Average Vina Docking Score (kcal/mol, ↓) | | | | |
| --- | --- | --- | --- | --- | --- | --- |
| | | ADRB2 | ALDH1 | ESR_ago | ESR_antago | FEN1 |
| Fragment | FragGFN | -10.19 (± 0.33) | -10.43 (± 0.29) | -9.81 (± 0.09) | -9.85 (± 0.13) | -7.67 (± 0.71) |
| | FragGFN+SA | -9.70 (± 0.61) | -9.83 (± 0.65) | -9.27 (± 0.95) | -10.06 (± 0.30) | -7.26 (± 0.10) |
| Reaction | SynNet | -8.03 (± 0.26) | -8.81 (± 0.21) | -8.88 (± 0.13) | -8.52 (± 0.16) | -6.36 (± 0.09) |
| | BBAR | -9.95 (± 0.04) | -10.06 (± 0.14) | -9.97 (± 0.03) | -9.92 (± 0.05) | -6.84 (± 0.07) |
| | SynFlowNet | -10.85 (± 0.10) | -10.69 (± 0.09) | -10.44 (± 0.05) | -10.27 (± 0.04) | -7.47 (± 0.02) |
| | RGFN | -9.84 (± 0.21) | -9.93 (± 0.11) | -9.99 (± 0.11) | -9.72 (± 0.14) | -6.92 (± 0.06) |
| | **RXNFLOW** | **-11.45** (± 0.05) | **-11.26** (± 0.07) | **-11.15** (± 0.02) | **-10.77** (± 0.04) | **-7.66** (± 0.02) |

Table 3: **Synthesizability.** Mean and standard deviation over 4 runs. The best results are in bold.

| Category | Method | AiZynthFinder Success Rate (%, ↑) | | | | |
| --- | --- | --- | --- | --- | --- | --- |
| | | ADRB2 | ALDH1 | ESR_ago | ESR_antago | FEN1 |
| Fragment | FragGFN | 4.00 (± 3.54) | 3.75 (± 1.92) | 1.00 (± 1.00) | 3.75 (± 1.92) | 0.25 (± 0.43) |
| | FragGFN+SA | 5.75 (± 1.48) | 6.00 (± 2.55) | 4.00 (± 2.24) | 1.00 (± 0.00) | 0.00 (± 0.00) |
| Reaction | SynNet | 54.17 (± 7.22) | 50.00 (± 0.00) | 50.00 (± 0.00) | 25.00 (± 25.00) | 50.00 (± 0.00) |
| | BBAR | 21.25 (± 5.36) | 19.50 (± 3.20) | 17.50 (± 1.50) | 19.50 (± 3.64) | 20.00 (± 2.12) |
| | SynFlowNet | 52.75 (± 1.09) | 57.00 (± 6.04) | 53.75 (± 9.52) | 56.50 (± 2.29) | 53.00 (± 8.92) |
| | RGFN | 46.75 (± 6.86) | 47.50 (± 4.06) | 50.25 (± 2.17) | 49.25 (± 4.38) | 48.50 (± 6.58) |
| | **RXNFLOW** | **60.25** (± 3.77) | **63.25** (± 3.11) | **71.25** (± 4.15) | **66.50** (± 4.03) | **65.50** (± 4.09) |

Table 4: **Synthetic complexity**. Mean and standard deviation over 4 runs. The best results are in bold.

| Category | Method | Average Number of Synthesis Steps (↓) | | | | |
| --- | --- | --- | --- | --- | --- | --- |
| | | ADRB2 | ALDH1 | ESR_ago | ESR_antago | FEN1 |
| Fragment | FragGFN | 3.60 (± 0.10) | 3.83 (± 0.08) | 3.76 (± 0.20) | 3.76 (± 0.16) | 3.34 (± 0.18) |
| | FragGFN+SA | 3.73 (± 0.09) | 3.66 (± 0.04) | 3.66 (± 0.07) | 3.67 (± 0.21) | 3.79 (± 0.19) |
| Reaction | SynNet | 3.29 (± 0.36) | 3.50 (± 0.00) | 3.00 (± 0.00) | 4.13 (± 0.89) | 3.50 (± 0.00) |
| | BBAR | 3.60 (± 0.17) | 3.62 (± 0.19) | 3.76 (± 0.04) | 3.72 (± 0.11) | 3.59 (± 0.14) |
| | SynFlowNet | 2.64 (± 0.07) | 2.48 (± 0.07) | 2.60 (± 0.25) | 2.45 (± 0.09) | 2.56 (± 0.29) |
| | RGFN | 2.88 (± 0.21) | 2.65 (± 0.09) | 2.78 (± 0.19) | 2.91 (± 0.23) | 2.76 (± 0.17) |
| | **RXNFLOW** | **2.42** (± 0.23) | **2.19** (± 0.12) | **1.95** (± 0.20) | **2.15** (± 0.18) | **2.23** (± 0.18) |

GFN+SA does not show meaningful improvement in synthesizability, implying that optimization of a cheap synthesizability estimation function is suboptimal.

Furthermore, it is noteworthy that RXNFLOW outperformed FragGFN which does not consider synthesizability. This improvement can be attributed to two key factors. First, the Enamine building block library is specifically curated for drug discovery, limiting the search space $\mathcal{X}$ to a drug-like chemical space and simplifying the optimization complexity. Second, RXNFLOW needs shorter trajectories compared to FragGFNs since it assembles molecules with large building blocks instead of atoms or small fragments. This is beneficial for trajectory balance objectives where stochastic gradient variance tends to increase over longer trajectories (Malkin et al., 2022).

Table 5: **CrossDocked2020 benchmark**. We report the average and median values over the average properties for each test pocket. The best results are in bold and the second ones are in underlined. We denote the Generation Success as Succ., Synthesizability as Synthesiz., and Diversity as Div. Reference means the known binding active molecules in the test set. MolCRAFT-large (Qu et al., 2024) is the result when generating with more atoms than the reference ligands.

| Category | Model | Succ. (↑) Avg. | Vina (↓) Avg. | Vina (↓) Med. | QED (↑) Avg. | QED (↑) Med. | Synthesiz. (↑) Avg. | Synthesiz. (↑) Med. | Div. (↑) Avg. | Time (↓) Avg. |
|---|---|---|---|---|---|---|---|---|---|---|
| Reference | - | - | -7.71 | -7.80 | 0.48 | 0.47 | 36.1% | - | - | - |
| Atom | Pocket2Mol | 98.3% | -7.60 | -7.16 | 0.57 | 0.58 | 29.1% | 22.0% | 0.83 | 2504 |
| | TargetDiff | 91.5% | -7.37 | -7.56 | 0.49 | 0.49 | 9.9% | 3.2% | 0.87 | 3428 |
| | DecompDiff | 66.0% | -8.35 | -8.25 | 0.37 | 0.35 | 0.9% | 0.0% | 0.84 | 6189 |
| | DiffSBDD | 76.0% | -6.95 | -7.10 | 0.47 | 0.48 | 2.9% | 2.0% | **0.88** | 135 |
| | MolCRAFT | 96.7% | -8.05 | -8.05 | 0.50 | 0.50 | 16.5% | 9.1% | 0.84 | 141 |
| | MolCRAFT-large | 70.8% | **-9.25** | **-9.24** | 0.45 | 0.44 | 3.9% | 0.0% | 0.82 | >141 |
| Fragment | TacoGFN | **100.0%** | -8.24 | -8.44 | **0.67** | **0.67** | 1.3% | 1.0% | 0.67 | **4** |
| Reaction | **RXNFLOW** | **100.0%** | -8.85 | -9.03 | **0.67** | **0.67** | 34.8% | 34.5% | 0.81 | **4** |

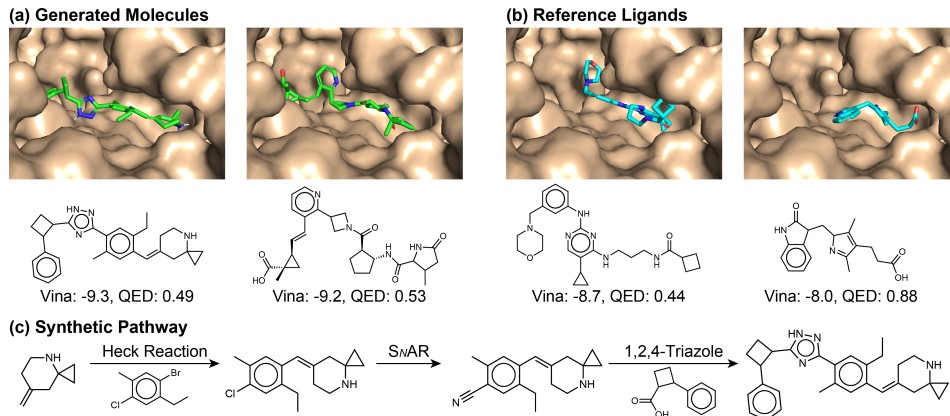

Figure 3: **Visualization of generated molecules** in a zero-shot manner. **(a-b)** Docking results of generated molecules and known reference ligands of TBK1 (PDB Id: 1FV, SU6). **(c)** Generative trajectory, which is the generated synthetic pathway of the left molecule in (a).

## 4.2 ZERO-SHOT SAMPLING VIA POCKET CONDITIONING

**Setup.** We extend our works to a pocket-conditional generation problem to design binders for arbitrary pockets without additional training oracles (Ragoza et al., 2022; Liu et al., 2022; Peng et al., 2022; Guan et al., 2023a;b; Schneuing et al., 2023). To address this challenge, we follow TacoGFN (Shen et al., 2023), which is a fragment-based GFlowNet for pocket-conditional generation. Since it requires more training oracles to learn pocket-conditional policies than target-specific generation, TacoGFN used pre-trained proxies that predict docking scores against arbitrary pockets using pharmacophore representation (Seo & Kim, 2024). Since RXNFLOW explicitly considers synthesizability, we exclude the SA score from the TacoGFN's reward function as described in Sec. C.2.

We generate 100 molecules for each of the 100 test pockets in the CrossDocked2020 benchmark (Francoeur et al., 2020) and evaluate them with the following metrics: **Vina (kcal/mol)** measures the average docking score from QuickVina (Alhossary et al., 2015). **QED** measures the average druglikeness of molecules. **Synthesizability (%)** is the fraction of synthesizable molecules. **Diversity** measures an average pairwise Tanimoto distance of ECFP4 fingerprints. Moreover, we report the **Generation Success (%)** which is the percentage of unique RDKit-readable molecules without disconnected parts, and **Time (sec.)** which is the average sampling time to generate 100 molecules.

**Baselines.** We compare RXNFLOW with state-of-the-art distribution learning-based models trained on a synthesizable drug set: an autoregressive model (**Pocket2Mol** (Peng et al., 2022)), diffusion models (**TargetDiff** (Guan et al., 2023a), **DiffSBDD** (Schneuing et al., 2023), **DecompDiff** (Guan et al., 2023b)), and bayesian flow network (**MolCRAFT** (Qu et al., 2024)). We also perform a comparison with an optimization-based **TacoGFN** (Shen et al., 2023). For a fair comparison with the distribution learning-based approaches, we used the docking proxy trained on CrossDocked2020.

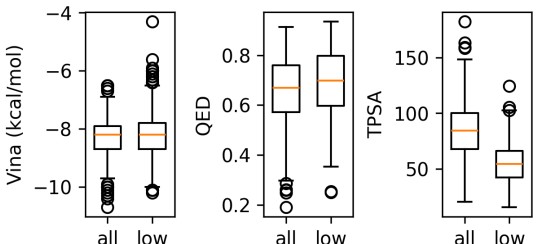
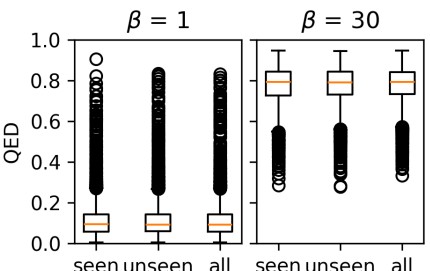

Figure 4: **Property distribution** of sampled molecules with "all" building blocks and "low"-TPSA building blocks. Vina score was calculated against the KRAS-G12C target.

Figure 5: **QED reward distribution** of generated molecules for each of the "seen", "unseen", and "all" blocks. Additional results are in Figure 14.

**Result.** As shown in Table 5, RXNFLOW achieves significant improvements in drug-related properties. In particular, RXNFLOW outperforms the docking score for TacoGFN and attains high drug-likeness while showing a competitive docking score with the state-of-the-art model, MolCRAFT-large. Moreover, RXNFLOW ensures the synthesizability comparable to known active ligands, outperforming the fragment-based TacoGFN trained on the SA score objective and the distributional learning-based models trained on synthesizable drug molecules. Figure 3 illustrates generated molecules against TANK-binding kinase 1 (TBK1) which is not included in the training set.

An important finding is that RXNFLOW maintains high structural diversity (0.81) despite the typical trade-off between optimization power and diversity (Gao et al., 2022a). This is a significant improvement over the fragment-based TacoGFN, which scored 0.67 and is comparable to the distributional learning-based models that range from 0.83 to 0.87. We attribute this enhancement to our action space, which contains chemically diverse building blocks, in contrast to the small and limited fragment sets used in fragment-based GFlowNets. This suggests that our model can effectively balance the potency and diversity of generated molecules.

### 4.3 INTRODUCING ADDITIONAL OBJECTIVE WITHOUT RETRAINING

In drug discovery, new objectives often arise during the research process, such as enhancing solubility, reducing toxicity, or improving selectivity (Fink et al., 2022; Joshi et al., 2021). These additional objectives typically not only require retraining models but also increase the optimization complexity. In this context, RXNFLOW can achieve some additional objectives by simply introducing constraints to MDP without retraining thanks to the non-hierarchical action space (Sec. 3.4). As shown in Figure 4, we explore the scenario of adding a solubility objective to a pre-trained GFlowNet in Sec. 4.2. Specifically, we target the generation of hydrophobic molecules by restricting the building blocks with topological polar surface area (TPSA) in the bottom 15% ("low") and sampled 500 molecules for the KRAS-G12C mutant (PDB Id: 6oim). While there are slight differences in the QED distributions due to the correlation between TPSA and QED, the generated molecules are more hydrophobic and retain similar overall reward distributions. We also performed the ablation study of non-hierarchical MDP in Sec. D.6.

### 4.4 SCALING ACTION SPACE WITHOUT RETRAINING

The building block libraries for drug discovery continue to grow, from 60,000 in 2020 to over 1.2 million blocks today, to enhance chemical diversity and novelty (Grygorenko et al., 2020). However, existing generative models require retraining to accommodate newly discovered building blocks, limiting their scalability and adaptability. On the other hand, RXNFLOW can integrate new building blocks without retraining by understanding the chemical context of actions through action embedding. We first divide the 1M-sized block library ("all") into two sets: 500,000 blocks for training ("seen") and the remaining blocks ("unseen"). After training with the QED objective on various reward exponent settings ($R^\beta$), we generate 5,000 molecules from each set ("seen", "unseen", and "all"). Figure 5 shows that the reward distributions of samples are nearly identical, demonstrating that RXNFLOW performs robustly with unseen building blocks. This result highlights the generalization ability and scalability of RXNFLOW, a significant advantage for real-world applications.

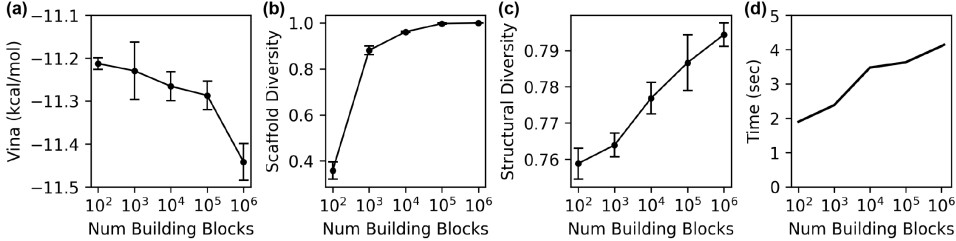

Figure 6: **Optimization power, diversity, and generation time** according to building block library size. **(a-c)** Average and standard deviation of properties of the top-1000 high-affinity molecules over 4 runs on pocket-specific generation. **(a)** Average docking score. **(b)** The uniqueness of Bemis-Murcko scaffolds. **(c)** Average Tanimoto distance. **(d)** Average runtime to generate 100 molecules over the CrossDocked2020 test pockets in a zero-shot manner.

### 4.5 ABLATION STUDY: THE EFFECT OF BUILDING BLOCK LIBRARY SIZE

Expanding the size of building block libraries provides an opportunity to discover more diverse and potent drug candidates (Grygorenko et al., 2020). In Sec. 4.1, RXNFLOW outperforms Syn-FlowNet and RGFN which use smaller block libraries, but differences in model architectures may have contributed to these results. To isolate the effect of the building block library size, we conduct an ablation study using partial libraries with a pocket-specific optimization task against the kappa-opioid receptor (PDB Id: 6b73), as illustrated in Figure 6(a-c). The results indicate that increasing the library size enhances both optimization power, in terms of docking scores, and diversity, in terms of a higher number of unique Bemis-Murcko scaffolds (Bemis & Murcko, 1996) and an increased Tanimoto diversity of the generated molecules. Additionally, as shown in Figure 6(d), the generation time only doubles on the 100-fold larger action space, highlighting the efficiency of RXNFLOW. These results demonstrate the forte of RXNFLOW in navigating a broader chemical space to discover novel drug candidates by overcoming the computational limitations for expanding the action space. We also investigated the scaling laws with other reaction-based GFlowNets in Sec. D.3.

### 4.6 TOWARDS FURTHER DEVELOPMENT

Our framework has room for improvement regarding more efficient learning and sampling. First, the 3D interaction modeling can enhance the generative performance. Given the high correlation between binding affinity and conformation, such spatial relationships could provide meaningful information for our generative model to make a reasonable decision. In Figure 7, we observed that the 3D interaction modeling using docking conformations boosts the discovery of candidate molecules against the beta-2-adrenergic receptor. Second, the current action space subsampling method forces exploration due to uniform sampling to minimize bias. We can enhance the exploitation by prioritizing the building blocks of action space subsampling instead of uniform subsampling.

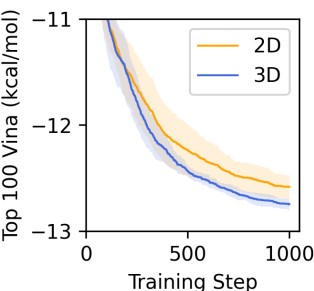

Figure 7: **Affect of 3D interaction modeling.** Mean and standard error over 3 seeds

## 5 CONCLUSION

In this work, we introduce RXNFLOW, a synthesis-oriented generative framework designed to explore broader chemical spaces, thereby enhancing both diversity and potency for drug discovery. Our framework efficiently handles massive action spaces to expand the search space without significant computational or memory overhead by employing a novel action space subsampling technique. RXNFLOW can effectively identify diverse drug candidates with desired properties and synthetic feasibility by learning the objective of generative flow networks on synthetic pathways. Additionally, by formulating a non-hierarchical MDP, RXNFLOW can model generative flows on modified action spaces, allowing it to achieve additional objectives and incorporate newly discovered building blocks without retraining. These results highlight the potential of RXNFLOW as a practical and versatile solution for real-world drug discovery.

ACKNOWLEDGEMENT

This work was supported by Basic Science Research Programs through the National Research Foundation of Korea (NRF), grant-funded by the Ministry of Science and ICT (NRF-2023R1A2C2004376, NRF-2022M3J6A1063021, RS-2023-00257479). In addition, this research was supported by a grant of Korean ARPA-H Project through the Korea Health Industry Development Institute (KHIDI), funded by the Ministry of Health & Welfare, Republic of Korea (RS-2024-00512498).

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

## A   THEORETICAL ANALYSIS

In this section, we provide the theoretical background of action subsampling. We define $\mathcal{U}$ as the uniform subsampling policy, i.e., $\mathcal{B}^* \sim \text{Uniform}(\{\mathcal{B}^* \subseteq \mathcal{B} \mid |\mathcal{B}^*| = k\})$.

**Bias of log forward policy.**   In this section, we prove that the log forward policy estimation is unbiased (Eq. (7)) with the following weights:

$$
w_a = \begin{cases} |\mathcal{B}|/|\mathcal{B}^*| & \text{if } a \in \mathcal{B} \\ |\mathcal{B}_r|/|\mathcal{B}_r^*| & \text{if } a \text{ is } (r,b) \text{ and } r \in \mathcal{R}_2 \\ 1 & \text{if } a \text{ is Stop or } a \in \mathcal{R}_1 \end{cases}
$$

For readability, we express the edge flow $F$ and forward policy $P_F$ where the action $a$ is $s \rightarrow s'$ as follows:

$$
F_\theta(s \rightarrow s'; \theta) = F_\theta(s, a; \theta), \quad P_F(s'|s; \theta) = P_F(a|s; \theta)
$$

Then, the forward policy $P_F$ and the estimated policy $\hat{P}_F$ (Eq. (7)) can be rewritten as follows:

$$
P_F(s'|s; \theta) = P_F(a|s; \theta) = \frac{F_\theta(s, a)}{F_\theta(s)} = \frac{F_\theta(s, a)}{\sum_{a' \in \mathcal{A}(s)} F_\theta(s, a')}
$$

$$
\hat{P}_F(s'|s; \mathcal{A}^*; \theta) = \hat{P}_F(a|s; \mathcal{A}^*; \theta) = \frac{F_\theta(s, a)}{\hat{F}_\theta(s; \mathcal{A}^*)} = \frac{F_\theta(s, a)}{\sum_{a' \in \mathcal{A}^*(s)} w_{a'} F_\theta(s, a')}
$$

The expectation of the estimated initial state flow $\hat{F}_\theta(s_0; \mathcal{A}^*) = \sum_{a \in \mathcal{A}^*} w_a F_\theta(s_0, a)$ is given by

$$
\mathbb{E}_{\mathcal{A}^* \sim \mathcal{P}(\mathcal{A})}[\hat{F}_\theta(s_0; \mathcal{A}^*)] = \mathbb{E}_{\mathcal{B}^* \sim \mathcal{U}(\mathcal{B})} \left[ \frac{|\mathcal{B}|}{|\mathcal{B}^*|} \sum_{b \in \mathcal{B}^*} F_\theta(s_0, b) \right] = \sum_{b \in \mathcal{B}} F_\theta(s_0, b) = F_\theta(s_0), \quad (13)
$$

For a later state $s \neq s_0$, the expectation of the state flow $\hat{F}_\theta(s; \mathcal{A}^*)$ is given by

$$
\begin{aligned}
\mathbb{E}_{\mathcal{A}^* \sim \mathcal{P}(\mathcal{A})}[\hat{F}_\theta(s; \mathcal{A}^*)] &= \sum_{a \in \mathcal{R}_1 \cup \{\text{Stop}\}} F_\theta(s, a) + \sum_{r \in \mathcal{R}_2} \mathbb{E}_{\mathcal{B}_r^* \sim \mathcal{U}(\mathcal{B}_r)} \left[ \frac{|\mathcal{B}_r|}{|\mathcal{B}_r^*|} \sum_{b \in \mathcal{B}_r^*} F_\theta(s, (r, b)) \right] \\
&= \sum_{a \in \mathcal{R}_1 \cup \{\text{Stop}\}} F_\theta(s, a) + \sum_{r \in \mathcal{R}_2} \sum_{b \in \mathcal{B}_r} F_\theta(s, (r, b)) \\
&= P_F(s'|s; \theta)
\end{aligned} \quad (14)
$$

**Variance of log forward policy.**   We define the standard deviation of the probability $P_F(-|s; \theta)$ as $\sigma_\theta(-|s)$. For the initial state $s_0$, the variance of $\log \hat{P}_F(s'|s_0; \mathcal{A}^*; \theta)$ is given by

$$
\begin{aligned}
& \text{Var}_{\mathcal{A}^* \sim \mathcal{P}(\mathcal{A})} \left[ \log \hat{P}_F(s'|s_0; \mathcal{A}^*; \theta) \right] \\
&= \text{Var}_{\mathcal{B}^* \sim \mathcal{U}(\mathcal{B})} \left[ \log F_\theta(s_0 \rightarrow s') - \log \left( \frac{|\mathcal{B}|}{|\mathcal{B}^*|} \sum_{b \in \mathcal{B}^*} F_\theta(s_0, b) \right) \right] \\
&= \text{Var}_{\mathcal{B}^* \sim \mathcal{U}(\mathcal{B})} \left[ \log \sum_{b \in \mathcal{B}^*} P_F(b|s_0; \theta) \right] \qquad \text{(by normalizing with } F_\theta(s_0)) \\
&\approx \frac{|\mathcal{B}|^2(|\mathcal{B}| - |\mathcal{B}^*|)}{|\mathcal{B}^*|(|\mathcal{B}| - 1)} \sigma_\theta(-|s_0)^2 \quad \text{(by Eq. (24))} \qquad (15)
\end{aligned}
$$

For the later state $s \neq s_0$, the variance of $\log \hat{P}_F(s'|s; \mathcal{A}^*; \theta)$ is:

$$\text{Var}_{\mathcal{A}^* \sim \mathcal{P}(\mathcal{A})} \left[ \log \hat{P}_F(s'|s; \mathcal{A}^*; \theta) \right]$$

$$= \text{Var}_{\mathcal{A}^* \sim \mathcal{P}(\mathcal{A})} \left[ \log \left( \sum_{a \in \mathcal{R}_1 \cup \{\texttt{Stop}\}} P_F(a|s; \theta) + \sum_{r \in \mathcal{R}_2} \frac{|\mathcal{B}_r|}{|\mathcal{B}_r^*|} \sum_{b \in \mathcal{B}_r^*} P_F((r,b)|s; \theta) \right) \right]$$

$$\approx \frac{\text{Var}_{\mathcal{A}^* \sim \mathcal{P}(\mathcal{A})} \left[ \cancel{\sum_{a \in \mathcal{R}_1 \cup \{\texttt{Stop}\}} P_F(a|s;\theta)} + \sum_{r \in \mathcal{R}_2} \frac{|\mathcal{B}_r|}{|\mathcal{B}_r^*|} \sum_{b \in \mathcal{B}_r^*} P_F((r,b)|s; \theta) \right]}{\mathbb{E}_{\mathcal{A}^* \sim \mathcal{P}(\mathcal{A})} \left[ \sum_{a \in \mathcal{R}_1 \cup \{\texttt{Stop}\}} P_F(a|s;\theta) + \sum_{r \in \mathcal{R}_2} \frac{|\mathcal{B}_r|}{|\mathcal{B}_r^*|} \sum_{b \in \mathcal{B}_r^*} P_F((r,b)|s; \theta) \right]^2}$$

$$= \frac{1}{\left( \cancel{\sum_{a \in \mathcal{A}(s)} P_F(a|s;\theta)} \right)^2} \text{Var}_{\mathcal{A}^* \sim \mathcal{P}(\mathcal{A}(s))} \left[ \sum_{r \in \mathcal{R}_2} \frac{|\mathcal{B}_r|}{|\mathcal{B}_r^*|} \sum_{b \in \mathcal{B}_r^*} P_F((r,b)|s; \theta) \right]$$

$$= \sum_{r \in \mathcal{R}_2} \frac{|\mathcal{B}_r|^2 (|\mathcal{B}_r| - |\mathcal{B}_r^*|)}{|\mathcal{B}_r^*|(|\mathcal{B}_r| - 1)} \sigma_\theta((r,-)|s)^2 \quad \text{(by Eq. (22))} \tag{16}$$

Finally, we get the variance of $\log \hat{P}_F(\tau; \theta)$ where $\tau = (s_0 \to ... \to s_n = x)$:

$$\text{Var}_{\mathcal{A}^* \sim \mathcal{P}(\mathcal{A})} \left[ \log \hat{P}_F(\tau; \mathcal{A}^*; \theta) \right]$$

$$\approx \frac{|\mathcal{B}|^2 (|\mathcal{B}| - |\mathcal{B}^*|)}{k|\mathcal{B}^*|(|\mathcal{B}| - 1)} \sigma_\theta(-|s_0)^2 + \sum_{t=1}^{n-1} \sum_{r \in \mathcal{R}_2} \frac{|\mathcal{B}_r|^2 (|\mathcal{B}_r| - |\mathcal{B}_r^*|)}{|\mathcal{B}_r^*|(|\mathcal{B}_r| - 1)} \sigma_\theta((r,-)|s_t)^2 \tag{17}$$

**Expectation of trajectory balance loss.** Due to the variance of forward policy, there is a bias of the trajectory balance loss equal to the variance of $\log \hat{P}_F(\tau)$:

$$\mathbb{E}_{\mathcal{A}^* \sim \mathcal{P}(\mathcal{A})}[\hat{\mathcal{L}}_{\text{TB}}(\tau)]$$

$$= \mathbb{E}_{\mathcal{A}^* \sim \mathcal{P}(\mathcal{A})} \left[ \log \frac{Z_\theta \hat{P}_F(\tau; \theta)}{R(x) P_B(\tau|x)} \right]^2 + \text{Var}_{\mathcal{A}^* \sim \mathcal{P}(\mathcal{A})} \left[ \log \frac{Z_\theta \hat{P}_F(\tau; \theta)}{R(x) P_B(\tau|x)} \right]$$

$$= \mathcal{L}_{\text{TB}}(\tau) + \text{Var}_{\mathcal{A}^* \sim \mathcal{P}(\mathcal{A})} \left[ \log \hat{P}_F(\tau; \theta) \right] \tag{18}$$

$$\approx \mathcal{L}_{\text{TB}}(\tau) + \frac{|\mathcal{B}|^2 (|\mathcal{B}| - |\mathcal{B}^*|)}{|\mathcal{B}^*|(|\mathcal{B}| - 1)} \sigma_\theta(-|s_0)^2 + \sum_{t=1}^{n-1} \sum_{r \in \mathcal{R}_2} \frac{|\mathcal{B}_r|^2 (|\mathcal{B}_r| - |\mathcal{B}_r^*|)}{|\mathcal{B}_r^*|(|\mathcal{B}_r| - 1)} \sigma_\theta((r,-)|s_t)^2 \tag{19}$$

Therefore, the gradient of trajectory balance loss is:

$$\mathbb{E}_{\mathcal{A}^* \sim \mathcal{P}(\mathcal{A})} \left[ \nabla_\theta \hat{\mathcal{L}}_{\text{TB}}(\tau) \right]$$

$$\approx \nabla_\theta \mathcal{L}_{\text{TB}}(\tau) + \frac{|\mathcal{B}|^2 (|\mathcal{B}| - |\mathcal{B}^*|)}{|\mathcal{B}^*|(|\mathcal{B}| - 1)} \nabla_\theta \sigma_\theta(-|s_0)^2 + \sum_{t=1}^{n-1} \sum_{r \in \mathcal{R}_2} \frac{|\mathcal{B}_r|^2 (|\mathcal{B}_r| - |\mathcal{B}_r^*|)}{|\mathcal{B}_r^*|(|\mathcal{B}_r| - 1)} \nabla_\theta \sigma_\theta((r,-)|s_t)^2 \tag{20}$$

Given that $\sigma_\theta(\cdot)$, which is the standard deviation of probability to select actions, is inversely proportional to $|\mathcal{B}|$ or $|\mathcal{B}_r|$, the bias is highly dependent on the subsampling size (e.g. $|\mathcal{B}^*|$) rather than the subsampling ratio (e.g. $|\mathcal{B}^*|/|\mathcal{B}|$). Moreover, we can decrease the bias by MC sampling for state flow estimation, and the bias is reversely proportional to the number of samples $k$. However, in the same computational cost (proportional to $\mathcal{O}(k \times \sum_r |\mathcal{B}_r^*|)$), using the larger partial action spaces without MC sampling is more precise than using smaller partial action spaces with multiple MC samples. In Sec. D.8, we experimentally show that the bias is relatively small to trajectory balance loss with the toy experiment.

### A.1 THEORETICAL BACKGROUNDS

**Variance of uniformly sampled subset**  For the set $\mathcal{X}$ with a size of $n$, we define its uniformly sampled subset with size $m$ as $\mathcal{X}' \sim \mathcal{P}(\mathcal{X})$. For the function $F(\mathcal{X}) = \sum_{x \in \mathcal{X}} f(x)$, we define the unbiased estimation:

$$\hat{F}(\mathcal{X}) = \frac{n}{m} F(\mathcal{X}') = \frac{n}{m} \sum_{x \in \mathcal{X}'} f(x)$$

When the mean and standard deviation of $f(x)$ is $\mu_f$ and $\sigma_f$, variance of $\hat{F}(\mathcal{X})$ is:

$$\text{Var}_{\mathcal{X}' \sim \mathcal{P}(\mathcal{X})} \left[ \hat{F}(\mathcal{X}) \right]$$

$$= \mathbb{E}_{\mathcal{X}' \sim \mathcal{P}(\mathcal{X})} \left[ \hat{F}(\mathcal{X})^2 \right] - \mathbb{E}_{\mathcal{X}' \sim \mathcal{P}(\mathcal{X})} \left[ \hat{F}(\mathcal{X}) \right]^2$$

$$= \frac{1}{{}_n C_m} \sum_{\mathcal{X}' \in \mathcal{P}(\mathcal{X})} \left( \frac{n}{m} \sum_{x \in \mathcal{X}'} f(x) \right)^2 - F(\mathcal{X})^2$$

$$= \frac{1}{{}_n C_m} \frac{n^2}{m^2} \left( {}_{n-1}C_{m-1} \sum_{i=1}^{n} f(b_i)^2 + {}_{n-2}C_{m-2} \sum_{i=1}^{n} \sum_{j>i}^{n} 2f(b_i)f(b_j) \right) - \left( \sum_{x \in \mathcal{X}} f(x) \right)^2$$

$$= \frac{n}{m} \sum_{i=1}^{n} f(x_i)^2 + \frac{2n(m-1)}{m(n-1)} \sum_{i=1}^{n} \sum_{j>i}^{n} f(x_i)f(x_j) - \left( \sum_{i=1}^{n} f(x_i)^2 + 2 \sum_{i=1}^{n} \sum_{j>i}^{n} f(x_i)f(x_j) \right)$$

$$= \frac{n-m}{m(n-1)} \left( (n-1) \sum_{i=1}^{n} f(x_i)^2 - 2 \sum_{i=1}^{n} \sum_{j>i}^{n} f(x_i)f(x_j) \right)$$

$$= \frac{n-m}{m(n-1)} \sum_{i=1}^{n} \sum_{j>i}^{n} (f(x_i) - f(x_j))^2$$

$$= \frac{n^2(n-m)}{m(n-1)} \sigma_f^2 \tag{21}$$

For MC sampling with $k$ samples, the variance is:

$$\text{Var}_{\mathcal{X}' \sim \mathcal{P}(\mathcal{X})} \left[ \hat{F}(\mathcal{X}) \right] = \frac{n^2(n-m)}{km(n-1)} \sigma_f^2 \tag{22}$$

The variance of $\log \hat{F}(\mathcal{X})$ is:

$$\text{Var}_{\mathcal{X}' \sim \mathcal{P}(\mathcal{X})} \left[ \log \hat{F}(\mathcal{X}) \right] \approx \frac{\text{Var}_{\mathcal{X}' \sim \mathcal{P}(\mathcal{X})} \left[ \hat{F}(\mathcal{X}) \right]}{\mathbb{E}_{\mathcal{X}' \sim \mathcal{P}(\mathcal{X})} \left[ \hat{F}(\mathcal{X}) \right]^2} = \frac{(n-m)}{km(n-1)} (\sigma_f / \mu_f)^2 \tag{23}$$

When the $F(\mathcal{X}) = 1$, the variance is:

$$\text{Var}_{\mathcal{X}' \sim \mathcal{P}(\mathcal{X})} \left[ \log \hat{F}(\mathcal{X}) \right] \approx \frac{m^2(n-m)}{km(n-1)} \sigma_f^2 \tag{24}$$

# B  RXNFLOW ARCHITECTURE

## B.1  FORWARD POLICY

We use the model architecture inspired from Cretu et al. (2024); Koziarski et al. (2024). We used a graph transformer (Yun et al., 2022) as the backbone $f_\theta$ following Bengio et al. (2021) and a multi-layer perceptron (MLP) for action embedding $g_\theta$. The graph embedding dimension is $d_1$, and the building block embedding dimension is $d_2$. The molecular graph for a state is $s$, and the GFlowNet condition vector is $c$ which includes a reward exponent and multi-objective optimization weights (Jain et al., 2023). $a\|b$ means the concatenation of two feature vectors $a$ and $b$.

**Initial block selection.**  For the first action, the model always selects AddFirstReactant action which selects $b \in \mathcal{B}$ for the starting molecule with $\text{MLP}_{\text{AddFirstReactant}} : \mathbb{R}^{d_1+d_2} \to \mathbb{R}$.

$$F_\theta(s_0, b, c) = \text{MLP}_{\text{AddFirstReactant}}(f_\theta(s_0, c)\|g_\theta(b)) \tag{25}$$

**Reaction selection.**  For the later states $s \neq s_0$, the model calculates the logits for the Stop action, ReactUni actions $r_1 \in \mathcal{R}_1$, and ReactBi actions $(r_2, b) \in \mathcal{R}_2 \times \mathcal{B}$.

The logit for Stop is calculated by $\text{MLP}_{\text{Stop}} : \mathbb{R}^{d_1} \to \mathbb{R}$:

$$F_\theta(s, \text{Stop}, c) = \text{MLP}_{\text{Stop}}(f_\theta(s, c)). \tag{26}$$

The logit for a ReactUni action $r_1 \in \mathcal{R}_1$ is calculated by $\text{MLP}_{\text{ReactUni}}^{r_1} : \mathbb{R}^{d_1} \to \mathbb{R}$:

$$F_\theta(s, r_1, c) = \text{MLP}_{\text{ReactUni}}^{r_1}(f_\theta(s, c)). \tag{27}$$

Finally, the logit for a ReactBi action $(r_2, b)$ is calculated by the one-hot embedding of reaction template $\delta(r_2) : \{0, 1\}^{|\mathcal{R}_2|}$ and $\text{MLP}_{\text{ReactBi}} : \mathbb{R}^{d_1+|\mathcal{R}_2|+d_2} \to \mathbb{R}$:

$$F_\theta(s, (r_2, b), c) = \text{MLP}_{\text{ReactBi}}(f_\theta(s, c)\|\delta(r_2)\|g_\theta(b)). \tag{28}$$

**Pocket conditioning.**  For a pocket-conditional generation, the model uses a $K$-NN pocket residual graph $\mathcal{G}^P$ and encodes GVP-GNN (Jing et al., 2020) according to Shen et al. (2023). The pocket conditions are included in the GFlowNet condition vector $c$.

**3D interaction modeling.**  Instead of using a 2D molecular graph of the ligand, the model can utilize a 3D binding complex graph to represent the state as the input of the graph transformer backbone $f_\theta$. In this graph, each ligand atom connects to its $K_1$ nearest protein atoms as well as to its neighboring ligand atoms, while each protein atom connects to its $K_2$ nearest protein atoms. The edges encode spatial relationship information between the connected nodes.

Furthermore, we lightweight the MDP formulation to implement computationally intensive 3D interaction modeling with modified layers $\text{MLP}_{\text{AddFirstReactant}}^* : \mathbb{R}^{d_1} \to \mathbb{R}^{d_2}$ and $\text{MLP}_{\text{ReactBi}}^* : \mathbb{R}^{d_1+|\mathcal{R}_2|} \to \mathbb{R}^{d_2}$ as follows:

$$F_\theta(s_0, b, c) = \text{MLP}_{\text{AddFirstReactant}}^*(f_\theta(s_0, c)) \odot g_\theta(b), \tag{29}$$

$$F_\theta(s, (r_2, b), c) = \text{MLP}_{\text{ReactBi}}^*(f_\theta(s, c)\|\delta(r_2)) \odot g_\theta(b), \tag{30}$$

## B.2  BACKWARD POLICY.

Malkin et al. (2022) introduced a uniform backward policy for small-scale drug discovery. However, in a directed acyclic graph (DAG) on synthetic pathways, where each state has numerous outgoing edges but few incoming ones, there is a significant imbalance in the number of trajectories along each incoming edge, depending on the distance from the initial state. In a uniform backward policy, where the flow of all incoming edges is equal, this imbalance diminishes the flow of trajectories that reach a state via shorter routes, i.e., those with fewer reaction steps. To facilitate shorter synthetic pathways as trajectories, we set the backward transition probability proportional to the expected number of trajectories along each incoming edge of the state, $a : s'' \to s$:

$$P_B(s''|s) = P((s'' \to s) \in \tau | s \in \tau) = \frac{\sum_{\tau \in \mathcal{T}:\tau=(s_0 \to \dots \to s'' \to s)} |\mathcal{A}(s)|^{N-|\tau|}}{\sum_{\tau \in \mathcal{T}:\tau=(s_0 \to \dots \to s)} |\mathcal{A}(s)|^{N-|\tau|}} \tag{31}$$

where $N$ is the maximum length of trajectories.

### B.3 BUILDING BLOCK REPRESENTATION FOR ACTION EMBEDDING.

To represent building blocks, we used both physicochemical properties, chemical structural properties, and topological properties. For physicochemical properties, we used 8 molecular features: molecular weight, the number of atoms, the number of H-bond acceptors/donors (HBA, HBD), the number of aromatic/non-aromatic rings, LogP, and TPSA. For chemical properties, we used the MACCS fingerprint (Durant et al., 2002), which represents the composition of the chemical functional groups in the molecule. For topological information, we used the Morgan ECFP4 fingerprint (Morgan, 1965) with a dimension of 1024, which is widely used in fingerprint-based deep learning researches. All properties are calculated with RDKit (Landrum et al., 2013).

### B.4 GFLOWNET TRAINING.

The training algorithm with the trajectory balance objective is described at Algorithm 1. The notations are in Sec. 3.3.

---
**Algorithm 1** Training GFlowNets with action space subsampling

---
1: **input** Entire action space $\mathcal{A}$, Maximum trajectory length $N$
2: **repeat**
3:     Sample partial action spaces $\mathcal{A}_0^*, \mathcal{A}_1^*, ..., \mathcal{A}_{N-1}^*$ from subsampling policy $\mathcal{P}(\mathcal{A})$
4:     Sample trajectory $\tau$ from sampling policy $\pi_\theta(-|-; \mathcal{A}_{(-)}^*)$
5:     Update model $\theta \leftarrow \theta - \eta \nabla \hat{\mathcal{L}}_{\text{TB}}(\tau)$
6: **until** model converges

---

### B.5 COMPARISON BETWEEN THREE GFLOWNET METHODS

Table 6: Comparison with SynFlowNet and RGFN. If the method includes each technique, it is denoted by an ◯, otherwise by an ×. RGFN investigates scaling up to 64,000 building blocks, but their experimental validation and proof-of-principle implementations use only 350.

| Methods | Hierarchical | Action Embedding | ReactUni | Num Blocks | Max Reaction Steps |
|---|---|---|---|---|---|
| Enamine REAL | - | - | ◯ | >1M | 3 |
| RGFN | ◯ | ◯ | × | 350-64,000 | 4 |
| SynFlowNet(v2024.5) | ◯ | × | ◯ | 6,000 | 4 |
| SynFlowNet(v2024.10) | ◯ | ◯ | ◯ | 10,000-220,000 | 3/4 |
| RXNFLOW | × | ◯ | ◯ | >1M | 3 |

While RXNFLOW shares similar rules with SynFlowNet (Cretu et al., 2024) and RGFN (Koziarski et al., 2024), there are several major differences as described in Table 6. Since there are two versions of SynFlowNet, the version presented at the ICLR 2024 GEM workshop and the version updated on October 16, 2024, we refer to the updated version as SynFlowNet(v2024.10). For the benchmark studies, we use the workshop version of SynFlowNet(v2024.5).

**Backward Trajectory.** In contrast to the simple fragment-based or atom-based GFlowNet, some backward transitions do not reach the initial state $s_0$. While RGFN and SynFlowNet do not consider the invalid transitions, SynFlowNet(upd) and RXNFLOW address this issue. To prevent the invalid backward transition, SynFlowNet(upd) introduces additional maximum likelihood and RE-INFORCE (Williams, 1992) training objectives for parameterized backward policy to prefer the backward transitions which can reach the initial state. In contrast, we implement explicit retrosynthetic analysis within maximum reaction steps for each state to collect valid backward trajectories.

**Flow Network Framework.** In contrast to SynFlowNet which uses discrete action space, RGFN, SynFlowNet(upd), and RXNFLOW include the action embedding (Dulac-Arnold et al., 2015) for chemical building blocks. While every building block must be sampled and trained at least once to prevent unsafe actions in a discrete action space, action embedding expresses actions based on structural information–domain knowledge. In particular, we formulate the non-hierarchical MDP instead of the hierarchical MDP for bi-molecular reactions. It is adaptable to modified building block libraries as described in Sec. B.6 but increases the computational cost and memory requirement dramatically. Thanks to action space subsampling, RXNFLOW can use non-hierarchical MDP without a computational burden problem.

**Sampling algorithm.** For each forward transition step, RGFN and SynFlowNet (both versions) model the flow for the entire action space to estimate the forward transition probability. Therefore, the computational cost and memory requirement is proportional to the number of using building blocks $\mathcal{B}$, i.e., $\mathcal{O}(|\mathcal{B}|)$. Instead of modeling all actions, we propose an action space subsampling technique, which is similar to the negative sampling technique (Mikolov et al., 2013) in natural language processing, to estimate the forward transition probability from the subset of action space. It makes a cost-variance trade-off to decrease the computational cost complexity to $\mathcal{O}(r|\mathcal{B}|)$, where $r$ is a subsampling ratio. As a result, RXNFLOW can handle millions of building blocks on training and inference with the computational cost of handling 10,000 building blocks.

### B.6    NON-HIERARCHICAL MARKOV DECISION PROCESS

To describe the difference, we supposed the situation that the toxicity is observed in molecules containing a functional group $-NR_3$, thereby excluding blocks with the function group from the building block library. In trained hierarchical MDP, the modification of the block library cannot change the probability of reaction templates, leading to the overestimation or underestimation of edge flows (the red color of Figure 8(a)). However, in the non-hierarchical MDP, the exclusion of some reactants does not affect actions that share the same reaction template (Figure 8(b)).

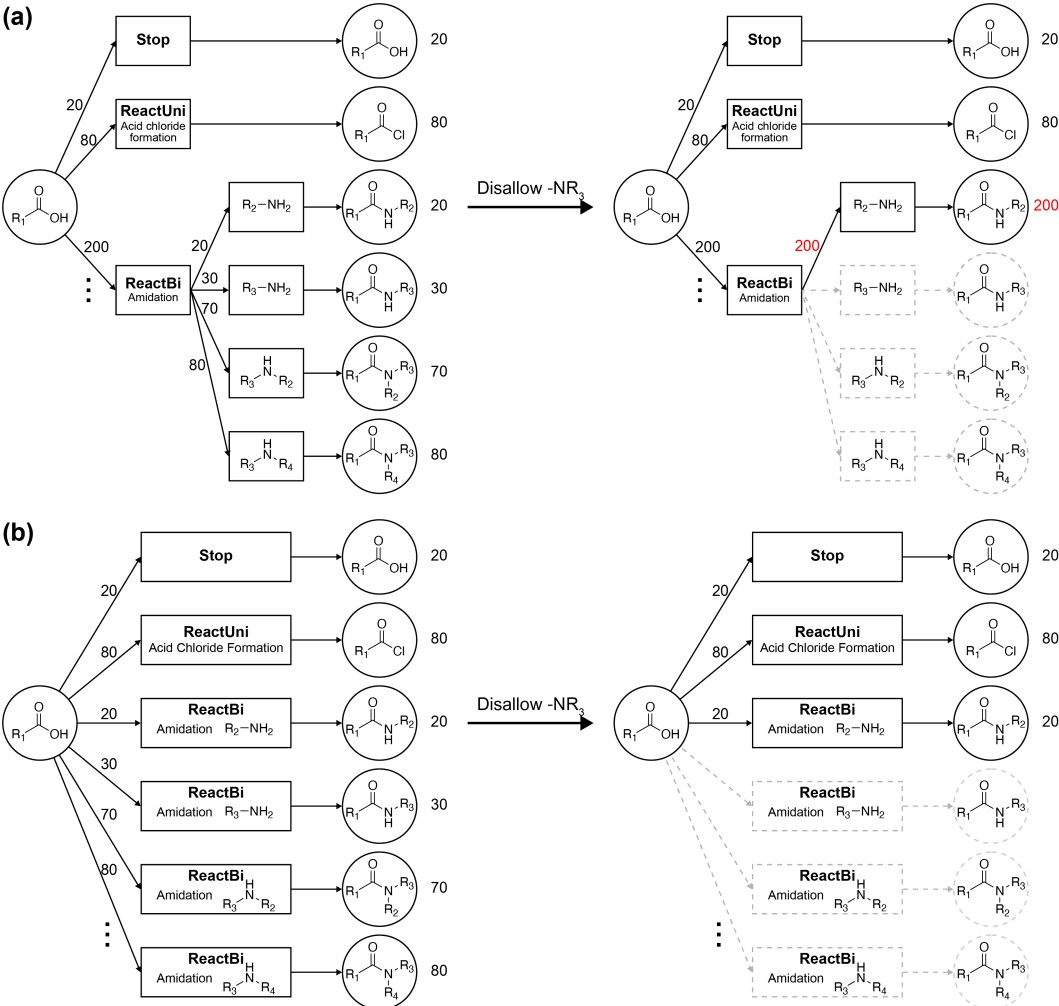

Figure 8: **Illustration of a situation** where an additional objective is introduced: excluding building blocks (reactants) containing the functional group $-NR_3$. The Gray dashed lines mean masked actions. **(a)** GFlowNet on hierarchical MDP. **(b)** GFlowNet on non-hierarchical MDP.

## C   EXPERIMENTAL DETAILS

### C.1   ACTION SPACE DETAILS

**Reaction templates.**   In this work, we used the reaction template set constructed by Cretu et al. (2024) from two public collections Hartenfeller et al. (2012); Button et al. (2019). The entire reaction template set includes 71 reaction templates, 13 for uni-molecular reactions and 58 for bi-molecular reactions. In *in silico* reactions with bi-molecular reaction templates, the products depend on the order of the input reactants. To ensure the consistency of the action, we consider two templates for each bi-molecular template according to the order of reactants, i.e. $|\mathcal{R}_1| = 13, |\mathcal{R}_2| = 116$. We note that our template set does not contain templates that have the same first and second reactant patterns.

**Building blocks.**   We used the Enamine comprehensive catalog with 1,309,385 building blocks released on 2024.06.10 (Grygorenko et al., 2020). We filtered out building blocks that are not RDKit-readable, have no possible reactions, or contain unallowable atoms[1], resulting in 1,193,871 remaining blocks.

### C.2   GFLOWNET TRAINING DETAILS

To minimize optimization performance influencing factors, we mostly followed the standard GFlowNet's model architecture and hyperparameters[2] except for some parameters in Table 7. All experiments were performed on a single NVIDIA RTX A4000 GPU.

Table 7: **Default hyperparameters** used in RXNFLOW training.

| Hyperparameters | Values |
| --- | --- |
| Minimum trajectory length | 2 (minimum reaction steps: 1) |
| Maximum trajectory length | 4 (maximum reaction steps: 3) |
| GFN temperature $\beta$ | Uniform(0, 64) |
| Train random action probability | 0.05 (5%) |
| Action space subsampling ratio | 1% |
| Building block embedding size | 64 |

For action space subsampling, we randomly subsample 1% actions for `AddFirstReactant` and each bi-molecular reaction template $r \in \mathcal{R}_2$. However, for bi-molecular reactions with small possible reactant block sets $\mathcal{B}_r \in \mathcal{B}$, the memory benefit from the action space subsampling is small while a variance penalty is large. Therefore, we set the minimum subsampling size to 100 for each bi-molecular reaction, and the action space subsampling is not performed when the number of actions is smaller than 100.

The number of actions for each action type is imbalanced, and the number of reactant blocks ($\mathcal{B}_r$) for each bi-molecular reaction template $r$ is also imbalanced. This can make some rare action categories not being sampled during training. We empirically found that `ReactBi` action were only sampled during 20,000 iterations (1.28M samples) in a toy experiment that uses one bi-molecular reaction template and 10,000 building blocks in some random seeds. Therefore, we set the random action probability as the default of 5%, and the model uniformly samples each action category in the random action sampling. This prevents incorrect predictions by ensuring that the model experiences trajectories including rare actions. We note that this random selection is only performed during model training.

### C.3   EXPERIMENTAL DETAILS

**Pocket-specific optimization.**   For pocket-specific optimization with GPU-accelerated UniDock, we normalize the Vina and SA scores for multi-objective optimization:

$$\widehat{\text{Vina}}(x) = -0.1 \max(\text{Vina}(x), 0), \quad \widehat{\text{SA}}(x) = (10 - \text{SA}(x))/9. \tag{32}$$

---

[1]The allowable atom types are B, C, N, O, F, P, S, Cl, Br, I.

[2]Default hyperparameters in https://github.com/recursionpharma/gflownet/blob/trunk/src/gflownet/tasks/seh_frag.py

For FragGFN, we set the maximum trajectory length to 9, and for SynFlowNet and RGFN, we used the same hyperparameters as our framework except for the maximum trajectory length and building block library size. According to their default setting, we set a the maximum trajectory length to 5 rather than 4, and we randomly sampled 350 building blocks for RGFN and 6000 building blocks for SynFlowNet. Moreover, we do not perform action subsampling for SynFlowNet and RGFN.

**Pocket-conditional generation in a zero-shot manner.** We used the modified version of the TacoGFN's reward function and training set. Since we don't need to optimize SA (Ertl & Schuffenhauer, 2009), we excluded the SA term from the reward functions:

$$r_{\text{affinity}}(x) = \begin{cases} 0 & \text{if } 0 \leq \text{Proxy}(x) \\ -0.04 \times \text{Proxy}(x) & \text{if } -8 \leq \text{Proxy}(x) \leq 0 \\ -0.2 \times \text{Proxy}(x) - 1.28 & \text{if } -13 \leq \text{Proxy}(x) \leq -8 \\ 1.32 & \text{if } \text{Proxy}(x) \leq -13 \end{cases}$$

$$r_{\text{QED}}(x) = \begin{cases} \text{QED}(x)/0.7 & \text{if } \text{QED}(x) \leq 0.7 \\ 1 & \text{otherwise} \end{cases}$$

$$r_{\text{SA}}(x) = \begin{cases} \widehat{\text{SA}}(x)/0.8 & \text{if } \widehat{\text{SA}}(x) \leq 0.8 \\ 1 & \text{otherwise} \end{cases}$$

$$\text{TacoGFN-Reward}(x) = \frac{r_{\text{affinity}}(x) \times r_{\text{QED}}(x) \times r_{\text{SA}}(x)}{\sqrt[3]{\text{HeavyAtomCounts}(x)}} \tag{33}$$

$$\text{RxnFlow-Reward}(x) = \frac{r_{\text{affinity}}(x) \times r_{\text{QED}}(x)}{\sqrt[3]{\text{HeavyAtomCounts}(x)}} \tag{34}$$

For hyperparameters, we set the pocket embedding dimension to 128 and the training GFN temperature to Uniform(0, 64) which are used in TacoGFN. We trained the model with 40,000 oracles whereas TacoGFN is trained for 50,000 oracles.

**Introducing further objectives without retraining.** For the restricted block library (TPSA<30), we set the action space subsampling ratio as 10% for both `AddFirstReactant` and `ReactBi`, and we set that as 1% for an entire library.

**Scaling action space without retraining.** For action space subsampling, we set the subsampling ratio as 2% for both `AddFirstReactant` and `ReactBi` for the "seen" and "unseen" libraries. For the "all" library ("seen" + "unseen"), we set the subsampling ratio as 1%.

**Ablation study.** We set different subsampling ratios according to the building block library size. For 100-sized, 1k-sized, and 10k-sized libraries, we do not perform the action space subsampling. We set an action space subsampling ratio of 10% for a 100k-sized one and 1% for a 1M-sized one.

**Further improvement with 3D interaction modeling.** To better analyze the effects of interaction modeling, we limit the maximum reaction step to 1. Since the initial state is an empty graph, we can analyze the impact of interaction modeling on a single decision process for selecting a reaction to perform. To obtain the 3D binding conformation, we used GPU-accelerated UniDock (Yu et al., 2023) under the *fast* search mode. We used the same neural network structure for both 2D-based generation and 3D-based generation. For the reward setting, we used the Vina docking score as a reward function and filtered out molecules according to the Lipinski Rule.

## C.4 Softwares

**Molecular docking software.** For a fair comparison with the baseline model, we used UniDock (Yu et al., 2023) for target-specific generation and QuickVina 2.1 (Alhossary et al., 2015) for SBDD. The initial ligand conformer is generated with srETKDG3 (Wang et al., 2020) in RDKit (Landrum et al., 2013). For QuickVina, we converted the molecule format to pdbqt with OpenBabel (O'Boyle et al., 2011) and AutoDock Tools (Huey et al., 2012). To set up an exhaustive search, we set the search mode to *balance* for UniDock and the exhaustiveness to 8 for QuickVina. We kept the seed fixed at 1 throughout the ETKDG and the whole docking process.

**Docking proxy.** We used the QuickVina proxy proposed by Shen et al. (2023) which is implemented in PharmacoNet (Seo & Kim, 2024). We used a proxy model trained on the CrossDocked2020 training set rather than the model trained on the ZINCDock15M training set.

**Synthetic accessibility estimation.** To evaluate the synthetic accessibility of molecules, we used the retrosynthesis planning tool AiZynthFinder (Genheden et al., 2020). AiZynthFinder uses MCTS to find synthesis paths and estimate the number of steps, search time, success rate, and synthetically accessible score as metrics to indicate synthesis complexity or synthesizability.

## C.5 BASELINES

**SynNet.** For SynNet (Gao et al., 2022b), we perform multi-objective optimization with the following reward function:

$$R(x) = 0.5\text{QED}(x) + 0.5\widehat{\text{Vina}}(x). \tag{35}$$

According to the standard setting for optimization, we set the number of offspring to 512 and the number of oracles to 125. To use pre-trained models, we used SynNet's template set (91 templates) instead of a template set (71 templates) used in our work.

**RGFN.** Since the code of RGFN is not released, we reimplement the RGFN.

**BBAR.** Since BBAR (Seo et al., 2023) allows multi-conditional generation, we directly used QED and docking scores without any processing. We split 64,000 ZINC20 molecules according to the reported splitting of BBAR: 90% for the training set, 8% for the validation set, and 2% for the test set (in our case, the number of sampling molecules). We performed UniDock for training and validation set to prepare the label for the molecules. Since BBAR requires the desired property value, we used the average docking score of the top 100 diverse modes from our model.

**Pocket2Mol, TargetDiff, DecompDiff, TacoGFN.** We followed the reported generative setting to generate 100 molecules for each CrossDocked test pocket. We set the center of the pockets with the reference ligands in the CrossDocked2020 database. We reuse reported runtime in Shen et al. (2023), which is measured on NVIDIA A100 for Pocket2Mol, TargetDiff, and DecompDiff, and NVIDIA RTX3090 for TacoGFN.

**DiffSBDD, MolCRAFT.** We used the generated samples from their official GitHub repository.

## C.6 LIT-PCBA POCKETS

Table 8 describes the protein information used in pocket-specific optimization with UniDock, which is performed on Sec. 4.1.

Table 8: **The basic target information** of the LIT-PCBA dataset and PDB entry used in this work.

| Target | PDB Id | Target name |
|---|---|---|
| ADRB2 | 4ldo | Beta2 adrenoceptor |
| ALDH1 | 5l2m | Aldehyde dehydrogenase 1 |
| ESR_ago | 2p15 | Estrogen receptor $\alpha$ with agonist |
| ESR_antago | 2iok | Estrogen receptor $\alpha$ with antagonist |
| FEN1 | 5fv7 | FLAP Endonuclease 1 |
| GBA | 2v3d | Acid Beta-Glucocerebrosidase |
| IDH1 | 4umx | Isocitrate dehydrogenase 1 |
| KAT2A | 5h86 | Histone acetyltransferase KAT2A |
| MAPK1 | 4zzn | Mitogen-activated protein kinase 1 |
| MTORC1 | 4dri | PPIase domain of FKBP51, Rapamycin |
| OPRK1 | 6b73 | Kappa opioid receptor |
| PKM2 | 4jpg | Pyruvate kinase muscle isoform M1/M2 |
| PPARG | 5y2t | Peroxisome proliferator-activated receptor $\gamma$ |
| TP53 | 3zme | Cellular tumor antigen p53 |
| VDR | 3a2i | Vitamin D receptor |

# D   ADDITIONAL RESULTS

## D.1   ADDITIONAL RESULTS FOR POCKET-SPECIFIC GENERATION TASK

We reported the additional results of Sec. 4.1 for the remaining 10 pockets on the LIT-PCBA benchmark.

Table 9: **Hit ratio** (%). Average and standard deviation for 4 runs. The best results are in bold.

| Category | Method | Hit ratio (%, ↑) | | | | |
| --- | --- | --- | --- | --- | --- | --- |
| | | GBA | IDH1 | KAT2A | MAPK1 | MTORC1 |
| Fragment | FragGFN | 5.00 (± 4.24) | 4.50 (± 1.66) | 1.25 (± 0.83) | 0.75 (± 0.83) | 0.00 (± 0.00) |
| | FragGFN+SA | 3.00 (± 1.00) | 4.50 (± 4.97) | 1.50 (± 0.50) | 2.00 (± 1.73) | 0.00 (± 0.00) |
| Reaction | SynNet | 50.00 (± 0.00) | 50.00 (± 0.00) | 29.17 (± 18.16) | 37.50 (± 21.65) | 0.00 (± 0.00) |
| | BBAR | 17.75 (± 2.28) | 19.50 (± 1.50) | 18.75 (± 1.92) | 16.25 (± 3.49) | 0.00 (± 0.00) |
| | SynFlowNet | 58.00 (± 4.64) | 59.00 (± 4.06) | 55.50 (± 10.23) | 47.25 (± 6.61) | 0.00 (± 0.00) |
| | RGFN | 48.00 (± 1.22) | 43.00 (± 2.74) | 49.00 (± 1.22) | 38.00 (± 4.12) | 0.00 (± 0.00) |
| | RXNFLOW | **66.00** (± 1.58) | **64.00** (± 5.05) | **66.50** (± 2.06) | **63.00** (± 4.64) | 0.00 (± 0.00) |
| | | OPRK1 | PKM2 | PPARG | TP53 | VDR |
| Fragment | FragGFN | 0.50 (± 0.50) | 7.25 (± 1.92) | 0.75 (± 0.43) | 4.25 (± 1.64) | 0.00 (± 0.00) |
| | FragGFN+SA | 0.50 (± 0.87) | 4.50 (± 1.50) | 1.00 (± 0.71) | 2.25 (± 1.92) | 0.00 (± 0.00) |
| Reaction | SynNet | 0.00 (± 0.00) | 0.00 (± 0.00) | 33.33 (± 20.41) | 8.33 (± 14.43) | 0.00 (± 0.00) |
| | BBAR | 2.50 (± 1.12) | 20.00 (± 0.71) | 10.50 (± 2.69) | 14.00 (± 3.94) | 0.00 (± 0.00) |
| | SynFlowNet | 23.50 (± 5.94) | 50.75 (± 1.09) | 53.50 (± 5.68) | 55.50 (± 9.94) | 0.00 (± 0.00) |
| | RGFN | 2.50 (± 2.06) | 34.75 (± 6.57) | 29.00 (± 6.52) | 37.00 (± 6.60) | 0.00 (± 0.00) |
| | RXNFLOW | **72.25** (± 2.05) | **62.00** (± 3.24) | **65.50** (± 4.03) | **67.50** (± 2.96) | **1.75** (± 0.83) |

Table 10: **Vina**. Average and standard deviation for 4 runs. The best results are in bold.

| Category | Method | Average Vina Docking Score (kcal/mol, ↓) | | | | |
| --- | --- | --- | --- | --- | --- | --- |
| | | GBA | IDH1 | KAT2A | MAPK1 | MTORC1 |
| Fragment | FragGFN | -8.76 (± 0.46) | -9.91 (± 0.32) | -9.27 (± 0.20) | -8.93 (± 0.18) | -10.51 (± 0.31) |
| | FragGFN+SA | -8.92 (± 0.27) | -9.76 (± 0.64) | -9.14 (± 0.43) | -8.28 (± 0.40) | -10.14 (± 0.30) |
| Reaction | SynNet | -7.60 (± 0.09) | -8.74 (± 0.08) | -7.64 (± 0.38) | -7.33 (± 0.14) | -9.30 (± 0.45) |
| | BBAR | -8.70 (± 0.05) | -9.84 (± 0.09) | -8.54 (± 0.06) | -8.49 (± 0.07) | -10.07 (± 0.16) |
| | SynFlowNet | -9.27 (± 0.06) | -10.40 (± 0.08) | -9.41 (± 0.04) | -8.92 (± 0.05) | -10.84 (± 0.03) |
| | RGFN | -8.48 (± 0.06) | -9.49 (± 0.13) | -8.53 (± 0.11) | -8.22 (± 0.15) | -9.89 (± 0.06) |
| | RXNFLOW | **-9.62** (± 0.04) | **-10.95** (± 0.05) | **-9.73** (± 0.03) | **-9.30** (± 0.01) | **-11.39** (± 0.09) |
| | | OPRK1 | PKM2 | PPARG | TP53 | VDR |
| Fragment | FragGFN | -10.28 (± 0.15) | -11.24 (± 0.27) | -9.54 (± 0.12) | -7.90 (± 0.02) | -10.96 (± 0.06) |
| | FragGFN+SA | -9.58 (± 0.44) | -10.83 (± 0.34) | -9.19 (± 0.29) | -7.61 (± 0.27) | -10.66 (± 0.61) |
| Reaction | SynNet | -8.70 (± 0.36) | -9.55 (± 0.14) | -7.47 (± 0.34) | -5.34 (± 0.23) | -10.98 (± 0.57) |
| | BBAR | -9.84 (± 0.10) | -11.39 (± 0.08) | -8.69 (± 0.10) | -7.05 (± 0.09) | -11.07 (± 0.04) |
| | SynFlowNet | -10.34 (± 0.07) | -11.98 (± 0.12) | -9.40 (± 0.05) | -7.90 (± 0.10) | -11.62 (± 0.13) |
| | RGFN | -9.61 (± 0.11) | -10.96 (± 0.18) | -8.53 (± 0.07) | -7.07 (± 0.06) | -10.86 (± 0.11) |
| | RXNFLOW | **-10.84** (± 0.03) | **-12.53** (± 0.02) | **-9.73** (± 0.02) | **-8.09** (± 0.06) | **-12.30** (± 0.07) |

Table 11: **Synthesizability**. Average and standard deviation for 4 runs. The best results are in bold.

| Category | Method | AiZynthFinder Success Rate (%, ↑) | | | | |
|---|---|---|---|---|---|---|
| | | GBA | IDH1 | KAT2A | MAPK1 | MTORC1 |
| Fragment | FragGFN | 5.00 (± 4.24) | 4.50 (± 1.66) | 1.25 (± 0.83) | 0.75 (± 0.83) | 2.75 (± 1.30) |
| | FragGFN+SA | 3.00 (± 1.00) | 4.50 (± 4.97) | 1.50 (± 0.50) | 3.25 (± 1.48) | 3.50 (± 2.50) |
| Reaction | SynNet | 50.00 (± 0.00) | 50.00 (± 0.00) | 45.83 (± 27.32) | 50.00 (± 0.00) | 54.17 (± 7.22) |
| | BBAR | 17.75 (± 2.28) | 19.50 (± 1.50) | 18.75 (± 1.92) | 16.25 (± 3.49) | 18.75 (± 3.90) |
| | SynFlowNet | 58.00 (± 4.64) | 59.00 (± 4.06) | 55.50 (± 10.23) | 47.25 (± 6.61) | 57.00 (± 7.58) |
| | RGFN | 48.00 (± 1.22) | 43.00 (± 2.74) | 49.00 (± 1.22) | 42.00 (± 3.00) | 44.50 (± 4.03) |
| | RXNFLOW | **66.00** (± 1.58) | **64.00** (± 5.05) | **66.50** (± 2.06) | **63.00** (± 4.64) | **70.50** (± 2.87) |
| | | OPRK1 | PKM2 | PPARG | TP53 | VDR |
| Fragment | FragGFN | 2.50 (± 2.29) | 8.75 (± 3.11) | 0.75 (± 0.43) | 4.25 (± 1.64) | 3.50 (± 2.18) |
| | FragGFN+SA | 3.25 (± 1.79) | 9.75 (± 2.28) | 1.25 (± 1.09) | 2.25 (± 1.92) | 3.75 (± 2.77) |
| Reaction | SynNet | 54.17 (± 7.22) | 50.00 (± 0.00) | 54.17 (± 7.22) | 29.17 (± 18.16) | 45.83 (± 7.22) |
| | BBAR | 13.75 (± 3.11) | 20.00 (± 0.71) | 15.50 (± 2.29) | 18.50 (± 3.28) | 12.25 (± 3.34) |
| | SynFlowNet | 56.50 (± 7.63) | 50.75 (± 1.09) | 53.50 (± 5.68) | 55.50 (± 9.94) | 53.50 (± 1.80) |
| | RGFN | 48.00 (± 2.55) | 48.50 (± 3.20) | 47.00 (± 5.83) | 53.25 (± 3.63) | 46.50 (± 2.69) |
| | RXNFLOW | **72.25** (± 2.05) | 62.00 (± 3.24) | **65.50** (± 4.03) | **67.50** (± 2.96) | **66.75** (± 2.28) |

Table 12: **Synthetic complexity**. Average and standard deviation for 4 runs. The best results are in bold.

| Category | Method | Average Number of Synthesis Steps (↓) | | | | |
|---|---|---|---|---|---|---|
| | | GBA | IDH1 | KAT2A | MAPK1 | MTORC1 |
| Fragment | FragGFN | 3.94 (± 0.11) | 3.74 (± 0.10) | 3.78 (± 0.09) | 3.72 (± 0.18) | 3.84 (± 0.18) |
| | FragGFN+SA | 3.94 (± 0.15) | 3.84 (± 0.23) | 3.66 (± 0.18) | 3.69 (± 0.21) | 3.94 (± 0.08) |
| Reaction | SynNet | 3.38 (± 0.22) | 3.38 (± 0.22) | 3.46 (± 0.95) | 3.50 (± 0.00) | 3.29 (± 0.36) |
| | BBAR | 3.71 (± 0.12) | 3.68 (± 0.02) | 3.63 (± 0.05) | 3.73 (± 0.05) | 3.77 (± 0.09) |
| | SynFlowNet | 2.48 (± 0.18) | 2.61 (± 0.13) | 2.45 (± 0.37) | 2.81 (± 0.24) | 2.44 (± 0.27) |
| | RGFN | 2.77 (± 0.20) | 2.97 (± 0.15) | 2.78 (± 0.10) | 2.86 (± 0.19) | 2.92 (± 0.06) |
| | RXNFLOW | **2.10** (± 0.08) | **2.16** (± 0.11) | **2.29** (± 0.05) | **2.29** (± 0.11) | **2.05** (± 0.09) |
| | | OPRK1 | PKM2 | PPARG | TP53 | VDR |
| Fragment | FragGFN | 3.82 (± 0.13) | 3.71 (± 0.12) | 3.73 (± 0.24) | 3.73 (± 0.23) | 3.75 (± 0.06) |
| | FragGFN+SA | 3.62 (± 0.12) | 3.84 (± 0.21) | 3.71 (± 0.04) | 3.66 (± 0.05) | 3.67 (± 0.25) |
| Reaction | SynNet | 3.29 (± 0.36) | 3.50 (± 0.00) | 3.29 (± 0.36) | 3.67 (± 0.91) | 3.63 (± 0.22) |
| | BBAR | 3.70 (± 0.17) | 3.61 (± 0.05) | 3.72 (± 0.13) | 3.65 (± 0.05) | 3.77 (± 0.16) |
| | SynFlowNet | 2.49 (± 0.33) | 2.62 (± 0.10) | 2.56 (± 0.12) | 2.51 (± 0.27) | 2.55 (± 0.09) |
| | RGFN | 2.81 (± 0.12) | 2.82 (± 0.10) | 2.82 (± 0.18) | 2.64 (± 0.10) | 2.84 (± 0.18) |
| | RXNFLOW | **2.00** (± 0.09) | **2.34** (± 0.19) | **2.21** (± 0.06) | **2.12** (± 0.12) | **2.12** (± 0.12) |

## D.2 PROPERTY DISTRIBUTION FOR POCKET-SPECIFIC GENERATION TASK

We reported the property distribution of the generated molecules for FragGFN, SynFlowNet, RGFN, and RXNFLOW for each of 15 LIT-PCBA targets.

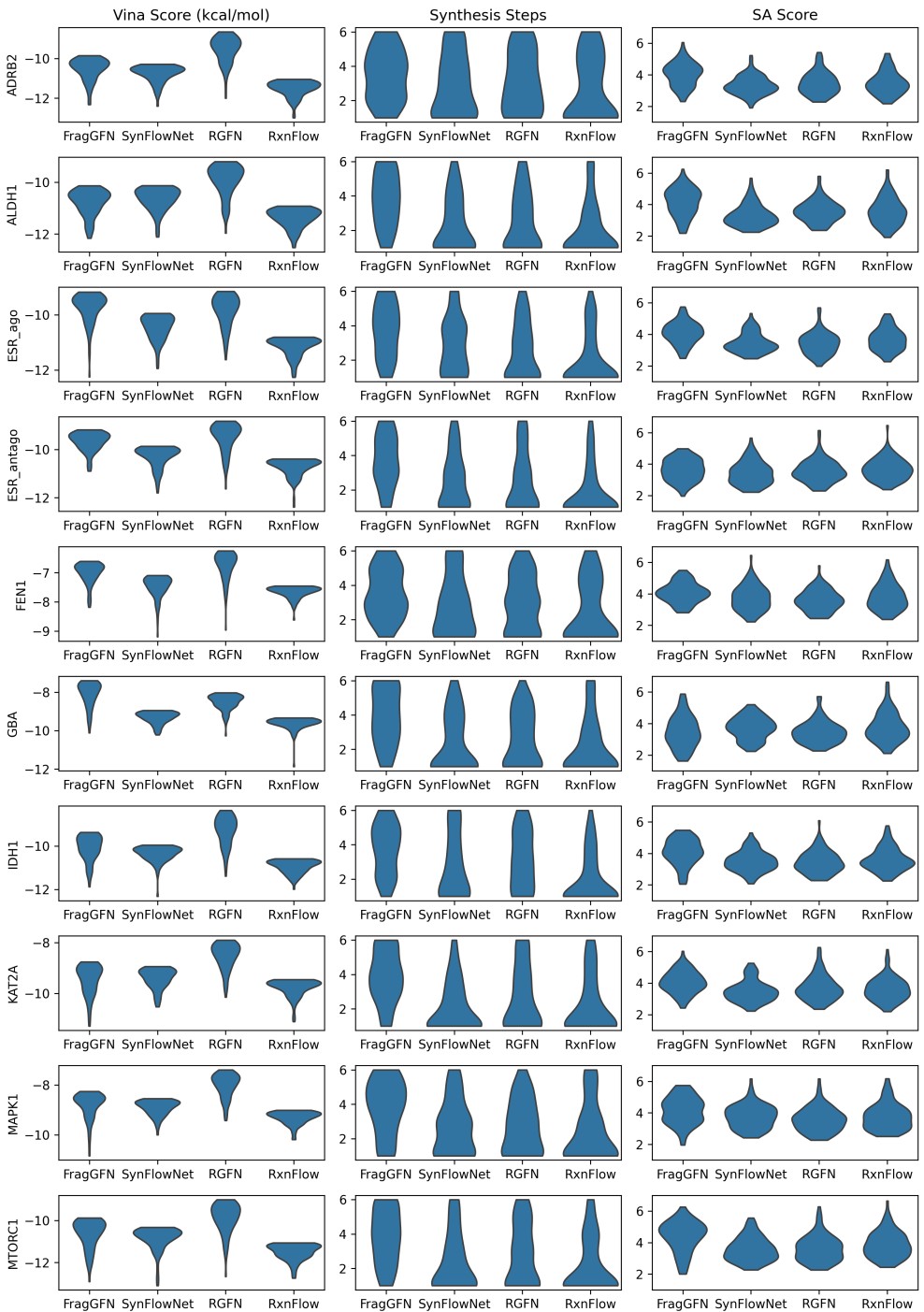

Figure 9: **The property distribution of the generated samples** for the first 10 LIT-PCBA targets.

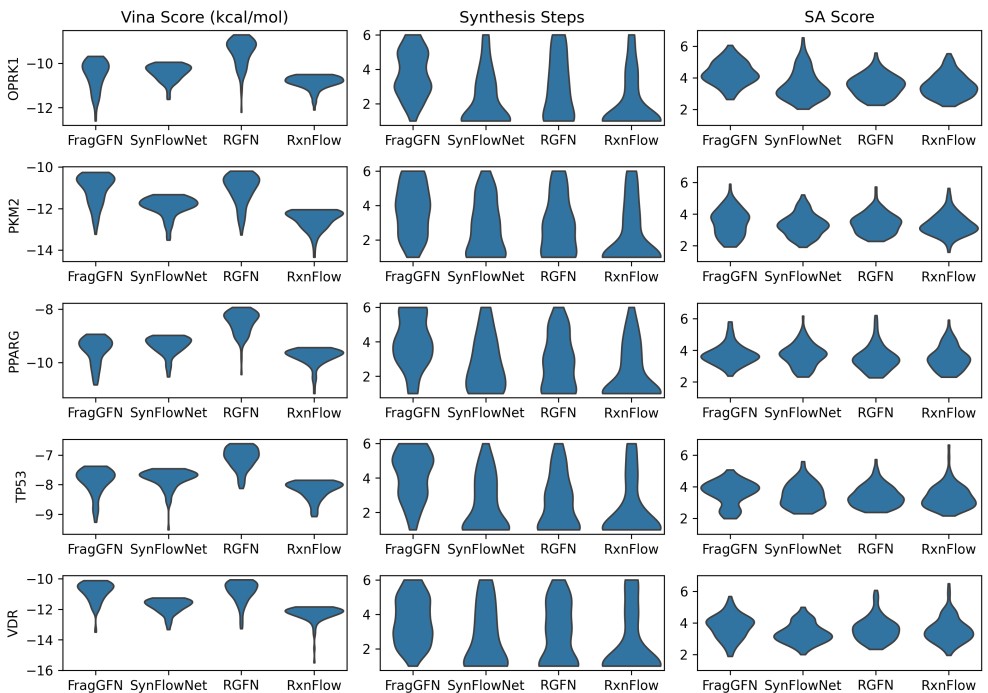

Figure 10: **The property distribution of the generated samples** for the last 5 LIT-PCBA targets.

### D.3 SCALING LAWS WITH BASELINE GFLOWNETS

In this section, we investigate the scaling laws of our model and the baseline GFlowNets, Syn-FlowNet and RGFN, focusing on performance (Figure 11) and computational cost (Figure 12). Our action space subsampling method reduces the computational cost and memory consumption via cost-variance trade-off and memory-variance trade-off.

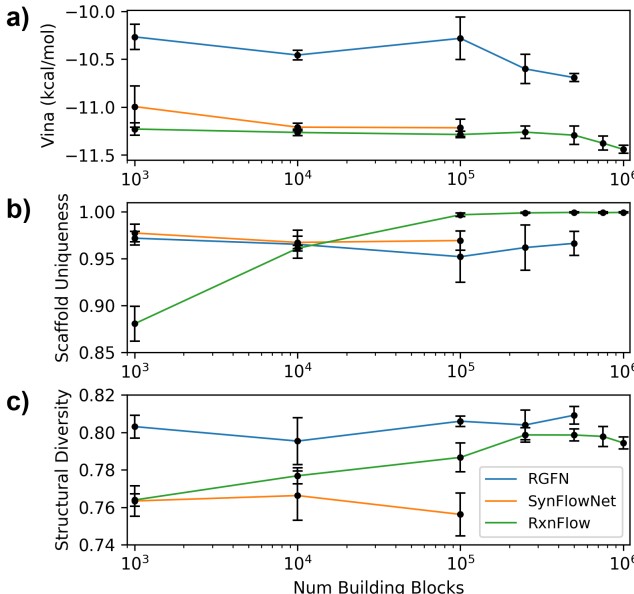

Figure 11: **Optimization power and diversity**. Average of standard deviation over the 4 runs. **(a)** Average docking score. **(b)** The uniqueness of Bemis-Murcko scaffolds. **(c)** Average Tanimoto distance.

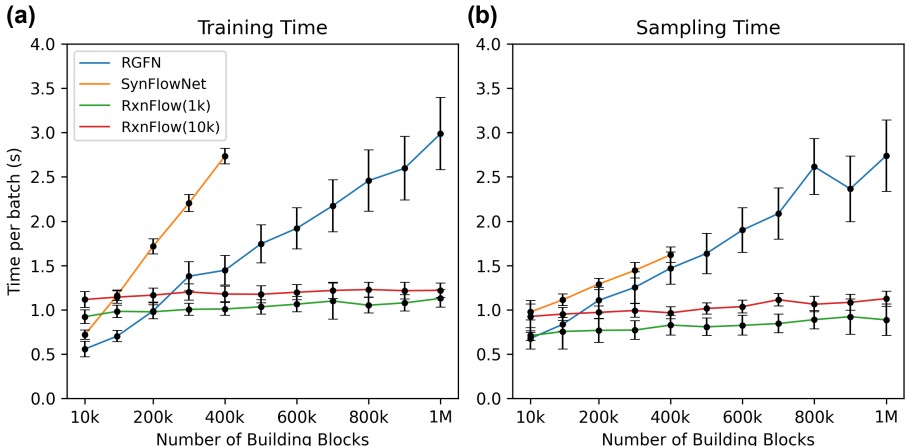

Figure 12: **Runtime** according to the building block library size. Average of standard deviation over the 100 batches. **(a)** The training runtime. **(b)** The sampling runtime without model training.

**Performance.** We conducted an optimization for the kappa-opioid receptor using RxNFLOW and baseline GFlowNets. To prevent the hacking of docking by increasing the molecular size, we performed Vina-QED multi-objective using the multi-objective GFlowNet framework (Jain et al., 2023). As shown in Figure 11, all three models exhibit similar trends in docking score optimization. However, conventional GFlowNets face memory resource constraints that scale with the action space size. As a result, SynFlowNet and RGFN were restricted to library sizes up to 100,000 and 500,000 sizes, respectively, due to memory limitations. In contrast, RxNFLOW can accommodate larger action spaces leveraging the memory-variance trade-off.

**Speed.** While SynFlowNet and RGFN consider all actions in the massive action space to estimate the forward transition probability, our architecture estimate Our architecture estimates forward transition probabilities over a subset of the action space, unlike SynFlowNet and RGFN, which compute all flows for the entire action space. This design allows RxNFLOW to handle larger action spaces without encountering computational bottlenecks. To evaluate the cost-efficiency of our technique, we measured runtime during both training and generation. To unify the generation environments across different methods, we used a constant reward function $R(x) = 1$ and a maximum reaction step of 1. Training times were measured on random trajectories, and sampling times were measured with initialized models, according to the building block library size. Due to memory limitations, SynFlowNet was restricted to library sizes up to 400,000.

For RxNFLOW, we tested two configurations: RxNFLOW (1k) and RxNFLOW (10k), which sample up to 1,000 and 10,000 building blocks, respectively. As demonstrated in Figure 12, RxNFLOW achieves significantly better cost efficiency in both training and generation compared to the baseline models. Additionally, computational costs can be easily reduced by adjusting the action-space subsampling ratio.

## D.4 STATISTICAL INFORMATION FOR POCKET-CONDITIONAL GENERATION TASK

Table 13: **Statistical Information for pocket-conditional generative models**. Mean and standard deviation for 5 sample sets.

| Category | Model | Vina (↓) Avg. | Vina (↓) Std. | QED (↑) Avg. | QED (↑) Std. |
|---|---|---|---|---|---|
| Atom | Pocket2Mol | -7.603 | 0.087 | 0.567 | 0.007 |
| | TargetDiff | -7.367 | 0.028 | 0.487 | 0.006 |
| | DiffSBDD | -6.949 | 0.079 | 0.467 | 0.002 |
| | DecompDiff | -8.350 | 0.033 | 0.368 | 0.004 |
| | MolCRAFT | -8.053 | 0.033 | 0.500 | 0.004 |
| | MolCRAFT-large | -9.302 | 0.033 | 0.448 | 0.002 |
| Fragment | TacoGFN | -8.237 | 0.268 | 0.671 | 0.002 |
| Reaction | **RXNFLOW** | -8.851 | 0.031 | 0.666 | 0.001 |

## D.5 TARGET SPECIFICITY OF GENERATED SAMPLES

To investigate the target specificity of pocket-conditional generation, we measured delta score (Gao et al., 2024a) for the top-10 molecules for each pocket. The delta score evaluates the pocket specificity of a proposed molecule by comparing the docking scores difference in how well each molecule binds to other proteins compared to the target protein.

Table 14: **Delta Score** for each methods

| Category | Model | Delta Score |
|---|---|---|
| Atom | Pocket2Mol | -1.74 |
| Atom | DecompDiff | -1.29 |
| Atom | DiffSBDD | -1.02 |
| Atom | MolCRAFT | -2.08 |
| Fragment | TacoGFN | -1.13 |
| Reaction | **RXNFLOW** | -1.13 |

## D.6 ABLATION STUDY FOR NON-HIERARCHICAL MARKOV DECISION PROCESS STRUCTURE

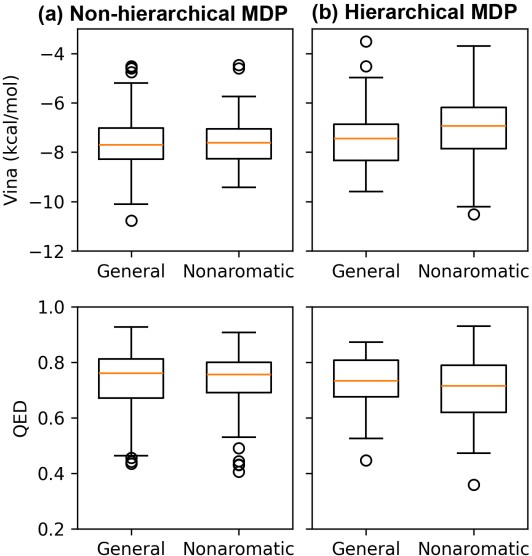

Figure 13: **Reward distribution of generated molecules** under different building block libraries: a randomly sampled building block (General) and a nonaromatic building block set (Nonaromatic). **(a)** The non-hierarchical MDP. **(b)** The hierarchical MDP.

To investigate the effectiveness of the non-hierarchical MDP structure, we performed an ablation study. We randomly selected 100,000 general building blocks from the Enamine building block library for model training (general). To simulate a scenario where specific functional groups are unallowed, we filtered out all aromatic building blocks to create a nonaromatic building block set. We trained the GFlowNets under Vina-QED multi-objective settings (Jain et al., 2023) against the beta-2 adrenergic receptor for 1,000 training oracles. After training, we generated 100 molecules using both the general block set and the nonaromatic block set without additional training.

As shown in Figure 13, the proposed non-hierarchical MDPs closely align with the identified reward distributions for both objectives, Vina and QED, on the general and nonaromatic building block sets. In contrast, hierarchical MDPs, as utilized in existing methodologies, demonstrate a shift in the reward distribution when the building block set is restricted. This indicates that non-hierarchical MDPs are more robust in changes in the building block set compared to hierarchical MDPs.

### D.7 ADDITIONAL RESULTS FOR SCALING ACTION SPACE WITHOUT RETRAINING

We reported the additional results for additional reward exponent settings ($R^\beta$).

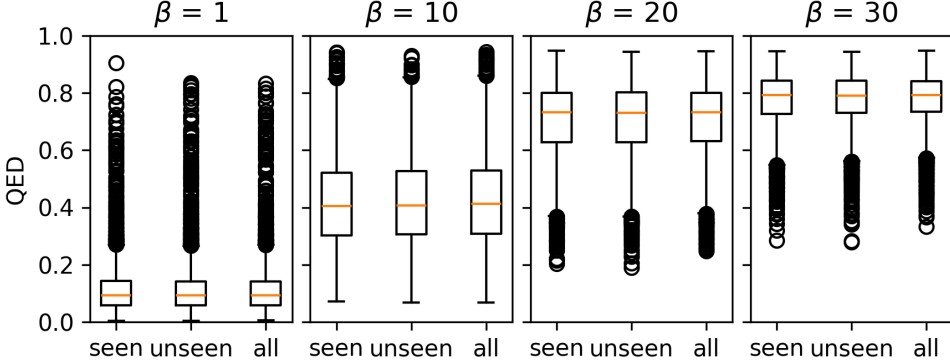

Figure 14: **QED reward distribution** of generated molecules.

Moreover, we investigated the generalization ability of our method to structurally different building blocks under the Vina-QED objectives against the beta-2 adrenergic receptor. For model training, we randomly selected 100,000 building blocks from the entire library (seen). Additionally, we selected 100,000 building blocks with a Tanimoto similarity of less than 0.5 to all training building blocks (unseen). The model was trained using the seen blocks first, and then the trained model subsequently generate molecules with seen blocks, unseen blocks, and a combination of both sets (all), respectively. As illustrated in Figure 15, the model can also generate samples with similar reward distributions from building blocks with different distributions than the ones used for training.

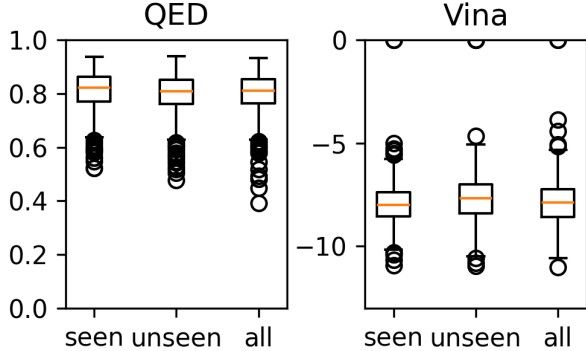

Figure 15: **Reward distribution** of generated molecules.

### D.8 THEORETICAL ANALYSIS

To assess the impact of action space subsampling in GFlowNet training, we conduct a toy experiment using a simplified setup with 10,000 blocks, one uni-molecular reaction template, one bi-molecular reaction template, and the QED objective. We used the Hell-Volhard-Zelinsky reaction as a uni-molecular reaction and the Amide reaction as a bi-molecular reaction, which are illustrated in Fig. 16. We used a minimum trajectory length of 1, max trajectory length of 2, constant GFN temperature of 1.0, and learning rate decay of 3,000 for $P_F$ and $\log Z$. For the GFlowNet sampler, we used the same weights of the proxy model, i.e. EMA factor of 0. We performed optimization for 30,000 oracles with a batch size of 64.

As shown in Figures 17(a) and 17(b), we compare a baseline GFlowNet trained without subsampling ("base") to models using various subsampling ratios and Monte Carlo (MC) sampling. The differences in $\log Z_\theta$ are relatively small ($<0.005$) across all settings, and increasing MC samples for state flow estimation $F_\theta$ further reduced the bias. In Figures 17(c) and 17(d), we also evaluate the bias in the trajectory balance loss ($\hat{\mathcal{L}}_{\mathrm{TB}}$) and its gradient norm ($\|\nabla_\theta \hat{\mathcal{L}}_{\mathrm{TB}}\|$) during training, finding negligible differences compared to the true values. These results indicate that our importance sampling reweighting approach effectively mitigates bias from action space subsampling, enabling efficient and accurate policy estimation.

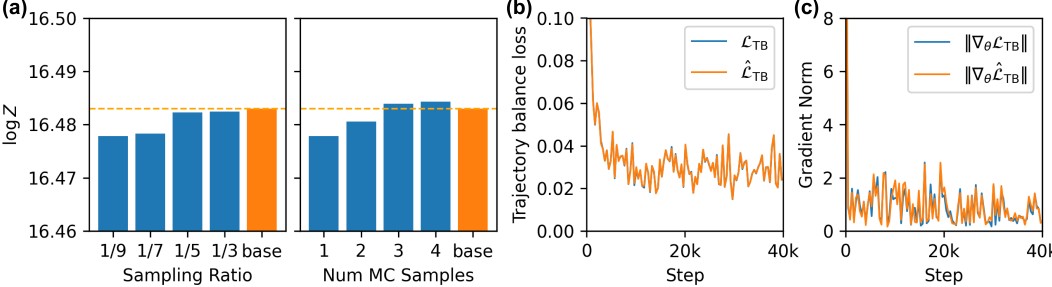

Figure 16: **Reaction templates employed in toy experiments**. **(a)** Hell-Volhard-Zelinsky reaction. **(b)** Amide reaction.

Figure 17: **Bias estimation.** **(a)** $\log Z_\theta$ according to the action space subsampling ratio (left) and the number of MC samples where the subsampling ratio is 1/9 (right). **(b)** The trajectory balance loss ($\mathcal{L}_{\mathrm{TB}}, \hat{\mathcal{L}}_{\mathrm{TB}}$) where the subsampling ratio is 1/9 under 4 MC samples. **(c)** The loss gradient norms ($\|\nabla_\theta \mathcal{L}_{\mathrm{TB}}\|, \|\nabla_\theta \hat{\mathcal{L}}_{\mathrm{TB}}\|$) where the subsampling ratio is 1/9 under 4 MC samples.

