# OpenReview forum: "Generative Flows on Synthetic Pathway for Drug Design"
_ICLR.cc/2025/Conference — ICLR 2025 Poster_

### Official Review · Reviewer_Q8DB · 2024-10-29

**Soundness:** 2
**Presentation:** 2
**Contribution:** 2
**Rating:** 5
**Confidence:** 3

**Summary:**

The author proposed a GFlowNet for on synthetic pathway, aiming to leverage the large action space for drug synthesis while retaining computational efficiency. The author proposed an action embedding which embeds the template and building blocks to parameterize the policy network. They further reduces the computational complexity by introducing a subsampling technique on the action space via importance sampling. The author validate their approach on LIT-PCBA, CrossDocked2020 dataset and outperformed baselines.

**Strengths:**

1. The proposed methods outperforms baseline in standard benchmark.
2. The methods exhibit nice scaling behavior on unseen blocks.

**Weaknesses:**

1. **Conceptual Novelty:** Utilizing reinforcement learning (RL) to determine reaction pathways is not a novel approach. The authors have simply applied this existing methodology to GFlowNets, which does not represent a significant conceptual advancement.

2. **Technical Novelty:** The proposed action embedding and importance sampling techniques do not appear to introduce new concepts. Essentially, the authors are reparameterizing the policy network with template and building block embeddings— a standard practice in RL when addressing large action spaces.

3. **Figure Clarity:** Figure 2 is currently confusing and requires further refinement. The use of diamonds to represent templates and squares for reactions may initially cause some misunderstanding. In contrast, Figure 7 is clearer and more effectively conveys the intended information.

4. **Efficiency Discussion:** The paper's main claim centers on reducing the action space to enhance computational tractability. However, there is a need for a more in-depth discussion and comparison of the training and inference times between the proposed network and existing baselines to substantiate this claim.

5. **Ablation Study:** The manuscript lacks an ablation study comparing Hierarchical Markov Decision Processes (MDPs) with non-hierarchical MDPs. Such a study is essential to evaluate the effectiveness and advantages of the non-hierarchical approach presented.

**Questions:**

1. Table 5, what is a and b?
2. line 45, what does it mean by 'chemical modification can degrade in the optimized propeties', can you give some examples?
3. It's unclear to me how you introduce the additional objective without retraining, can you clarity that?

---

> ### Author Response · Authors · 2024-11-22
> **Response to Reviewer Q8DB (1/n)**
>
> We appreciate the reviewer for valuable feedbacks on various aspects. Please see our response below:
>
> **Issue 1, 2: Conceptual Novelty and Technical Novelty**
>
> It is true that our approach builds on the well-structured reinforcement learning algorithm and an action embedding. We do not claim novelty for the use of these popular components.
>
> **However, we claim the contribution for the new scaling method of GFlowNet, action space subsampling, which can improve performance by simply expanding action spaces without additional computing resources.** Although the action embedding was already introduced in the GFlowNet in previous work (RGFN [1]), it doesn’t address the cost and memory on large action spaces. Since conventional GFlowNet requires computational cost and memory capacity proportional to the size of the action space, RGFN and SynFlowNet are slow in large behavior spaces and cannot handle all Enamine REAL building blocks due to memory limitations. On the other hand, our method dramatically reduces the amount of computation and enables training even in limited computing environments. To support our claim, we have included the scaling laws for GFlowNet performance and speed in Section D.3 in the revised manuscript.
>
> **Issue 3: Figure Clarity: "Figure 2 is currently confusing and requires further refinement. The use of diamonds to represent templates and squares for reactions may initially cause some misunderstanding."**
>
> We thank to reviewer for detailed feedback on Figure 2. We have modified Figure 2 more intuitively in the revised manuscript.
>
> **Issue 4: Efficiency Discussion: "The paper's main claim centers on reducing the action space to enhance computational tractability. However, there is a need for a more in-depth discussion and comparison of the training and inference times between the proposed network and existing baselines to substantiate this claim."**
>
> Thanks you for suggesting an additional experiments.  We have included both training time and generation time for ours and baseline models in the Section D.3 of the revised manuscript. While existing GFlowNets require computing cost proportional to the number of building blocks, our methodology allows GFlowNet to handle large building block library while maintaining the computing costs.
>
> The tables below include the average of runtime per batch for 100 batch (SynFlowNet has limited up to 400,000 building blocks due to a memory limitation.):
> 1. Training Time
> |    Num Blocks     | 10,000 | 100,000 | 200,000 | 300,000 | 400,000 | 500,000 | 600,000 | 700,000 | 800,000 | 900,000 | 1,000,000 |
> |----------------|--------|---------|---------|---------|---------|---------|---------|---------|---------|---------|-----------|
> | SynFlowNet    | 0.72   | 1.16    | 1.72    | 2.21    | 2.73    | -       | -       | -       | -       | -       | -         |
> | RGFN          | 0.56   | 0.70    | 0.99    | 1.38    | 1.45    | 1.74    | 1.92    | 2.17    | 2.46    | 2.60    | 2.99      |
> | RxnFlow       | 1.12   | 1.14    | 1.17    | 1.20    | 1.18    | 1.18    | 1.20    | 1.22    | 1.23    | 1.22    | 1.22      |
> 2. Sampling Time
> | Num Blocks          | 10,000 | 100,000 | 200,000 | 300,000 | 400,000 | 500,000 | 600,000 | 700,000 | 800,000 | 900,000 | 1,000,000 |
> |----------------|--------|---------|---------|---------|---------|---------|---------|---------|---------|---------|-----------|
> | SynFlowNet    | 0.98   | 1.11    | 1.29    | 1.44    | 1.62    | -       | -       | -       | -       | -       | -         |
> | RGFN          | 0.68   | 0.84    | 1.11    | 1.25    | 1.47    | 1.64    | 1.90    | 2.09    | 2.62    | 2.37    | 2.74      |
> | RxnFlow       | 0.93   | 0.95    | 0.97    | 0.99    | 0.97    | 1.01    | 1.04    | 1.11    | 1.06    | 1.08    | 1.13      |
>
> **Issue 5: Ablation Study: "The manuscript lacks an ablation study comparing Hierarchical Markov Decision Processes (MDPs) with non-hierarchical MDPs. Such a study is essential to evaluate the effectiveness and advantages of the non-hierarchical approach presented."**
>
> Thank you for pointing out the lack of an ablation study. We included the ablation study for the non-hierarchical MDP in Section D.6 in the revised manuscript.

---

> ### Author Response · Authors · 2024-11-22
> **Response to Reviewer Q8DB (2/n)**
>
> **Question 1: Table 5, "what is a and b?"**
>
> We thank the reviewer for pointing out the missing part. $a$ and $b$ mean the runtimes were measured on NVIDIA A100 and NVIDIA RTX 3090 respectively, while our model's runtime was measured on NVIDIA RTX A4000. Since our method’s speed was measured on the slower environment, we removed these superscripts and included it on the section C.5 in the revised manuscript.
>
> **Question 2: line 45, "what does it mean by 'chemical modification can degrade in the optimized propeties', can you give some examples?"**
>
> If the substructure of the molecule that is core structure of the objective property (e.g., creating an H-bond with a protein) is not synthetically accessible, modifying this substructure to a synthetically available structure may degrade the optimized property. We included an additional recent work [2] in the revised manuscript. As illustrated in Figure 3 of the reference [2], the optimized scores can be degraded during the modification.
>
> **Question 3: "It's unclear to me how you introduce the additional objective without retraining, can you clarify that?"**
>
> Sometimes, we may need to further tune the properties (e.g., cell permeability, molecular weight, toxicity) after the training of generative model is finished. In these cases, we can achieve these property constraints by simply modifying the building block library. For example, we can increase the cell permeability of generated molecules by simply using the low TPSA building blocks only, without model retraining.
>
> Of course, this can be addressed through multi-objective optimization by introducing additional property objectives. However, it is well-known that the complexity of multi-objective optimization increases significantly as the number of objectives grows. Furthermore, in practical drug discovery pipelines, binding affinity estimation often relies on computationally intensive docking tools such as GLIDE or GOLD, and the model training costs are high. For these cases, our proposed feature can be useful.
>
> ---
> **Reference:**
> 1. Koziarski, Michał, et al. "RGFN: Synthesizable Molecular Generation Using GFlowNets." arXiv preprint arXiv:2406.08506 (2024).
> 2. Gao, Wenhao, Shitong Luo, and Connor W. Coley. "Generative Artificial Intelligence for Navigating Synthesizable Chemical Space." arXiv preprint arXiv:2410.03494_ (2024).

---

> > ### Comment · Reviewer_Q8DB · 2024-11-23
> >
> > Thank you for your response. I ll maintain my score.

---

### Official Review · Reviewer_ovVP · 2024-10-30

**Soundness:** 3
**Presentation:** 3
**Contribution:** 3
**Rating:** 6
**Confidence:** 4

**Summary:**

In this manuscript, the authors adopted the GFlowNet to generate synthesizable molecules based on the protein pocket. They were able to demonstrate that RXNFLOW outperforms existing reaction-based and fragment-based models in pocket-specific synthetic pathways, and their method achieves state-of-the-art performance on CrossDocked2020 for pocket-conditional generation.

**Strengths:**

Overall, this paper is interesting. The GFlowNet model is suitable to be adopted to solve the synthesizability problem of small molecules. The problem with GFlowNet is that it lacks a method to construct a graph that covers the entire space of small molecules. This paper cleverly utilizes chemical synthesis reactions to construct this graph, while also adopting a down-sampling approach to enhance the effectiveness of the method.

**Weaknesses:**

The main issue is the insufficient characterization of 3D interactions; the model does not explicitly model the interactions between atoms in 3D space and only uses the generated molecule's vina score with the pocket as the reward. In addition, a careful discussion of the differences from SynFlowNet is needed. Currently, the authors have avoided mentioning SynFlowNet in the introduction. According to my understanding, the method proposed in this paper has no structural difference from SynFlowNet in terms of the model framework, with some improvements in sampling techniques. I hope the authors can discuss the differences from SynFlowNet in detail from the perspectives of graph construction for the flow, reward settings, and the sampling algorithm.

**Questions:**

1. For conventional SBDD methods, they are generally trained on molecules that can bind to protein pockets, resulting in generated small molecules with a specific distribution. I am curious about how the molecules generated in this space of synthesizable reaction-based small molecules differ from those generated by SBDD methods, such as pocket2mol. Because drug-likeness is not solely determined by synthesizability and vina score alone. For instance, the bond angle, number of rings. These facts are hard to encode into the reward function.
2. Please compare the speed of different methods.
3. Why use 3 reaction steps for your method and 4 steps for SynFlowNet and RGFN?

---

> ### Author Response · Authors · 2024-11-22
> **Response to Reviewer ovVP (1/n)**
>
> We appreciate the reviewer for valuable feedbacks. We answer each question one by one below:
>
> **Issue 1: "The main issue is the insufficient characterization of 3D interactions; the model does not explicitly model the interactions between atoms in 3D space and only uses the generated molecule's vina score with the pocket as the reward."**
>
> We fully agree that various 3d interaction modeling methods and our synthesis-oriented methods should be integrated. We included these points as future improvements in the revised manuscript.
>
> However, we would like to emphasize that **the primary goal of this work is not exploring better modeling methods for the protein-ligand interaction, but rather proposing a cost-efficient learning technology for scaling GFlowNet to massive synthetic action spaces.** Current GFlowNet architecture requires computing cost and memory capacity proportional to the size of action space. As demonstrated in Section D.3 in revised manuscript, existing GFlowNets face significant challenges when handling million-scale building block libraries. Therefore, this work has focused on how to address the computational burden of GFlowNet on large action spaces.
>
> **Issue 2-1:  "In addition, a careful discussion of the differences from SynFlowNet is needed. Currently, the authors have avoided mentioning SynFlowNet in the introduction. According to my understanding, the method proposed in this paper has no structural difference from SynFlowNet in terms of the model framework, with some improvements in sampling techniques."**
>
> We would like to clarify that **current SynFlowNet's manuscript (v2 in arXiv, released at 10/16) and code (released at 10/18) were updated after the ICLR submission deadline (10/1)**.
> Consequently, our manuscript is based on the previous version of SynFlowNet (v1, accepted in ICLR 2024 workshop).
> We made multiple improvements from the workshop version of SynFlowNet(v1) and it is also different in some parts to the updated SynFlowNet(v2).
>
> Given the timeline, we kindly request that our work be evaluated relative to the SynFlowNet(v1), the version available prior to the ICLR submission deadline.
>
> **Issue 2-2: "I hope the authors can discuss the differences from SynFlowNet in detail from the perspectives of graph construction for the flow, reward settings, and the sampling algorithm."**
>
> In the revised manuscript, we included the detailed comparison between our models and SynFlowNet (both v1, v2) in Section B.5. Here is a detailed comparison:
> - **Model Framework:** Compared to SynFlowNet(v1), the SynFlowNet(v2) and RxnFlow both employed the action embedding. Moreover, our method selects the reactions and building blocks at once (non-hierarchical MDP), whereas both versions of SynFlowNet selects the reactions first and then the building blocks (hierarchical MDP). We included the ablation study of this component in Section D.6.
> - **Reward setting:** The reward functions and optimization algorithms are independent; the optimization architectures can work on different reward functions. For fair comparison, we used the same reward function (multi-objective GFlowNet [1]) for training on all GFlowNet baselines in our comparative study (Table 1-4). We clarify this point in the revised manuscript.
> - **Preventing method for invalid backward trajectories:** An existence of invalid backward trajectory is a critical issue to achieving the GFlowNet objective. SynFlowNet(v1) does not consider this issue, but the updated SynFlowNet(v2) and ours address this issue in different ways. SynFlowNet(v2) introduces additional training objectives to prefer the backward transition which can reach the initial state, on the other hand, we implements explicit retrosynthetic analysis to collect the valid backward trajectories.
> - **Sampling algorithm:** The existing GFlowNets require computing cost proportional to the size of action spaces. Our proposed action space subsampling technique reduces both memory consumption and computing cost to scale GFlowNet on the large action space without additional computing resources through cost-variance trade-off. We have investigated the scaling laws for ours and baseline models in Section D.3.

---

> > ### Comment · Reviewer_ovVP · 2024-11-23
> >
> > To achieve the SBDD task, one need to model the interaction among atoms in the 3D space. The authors' response to this weakness is that they will leave this in the future work.  To compare with SynFlowNet, they require us to consider the improvement of this work in comparison to SynFlowNet v1 instead of SynFlowNet v2. I am not sure whether this is proper and I guess I will leave this question to the chair.
> >
> > Regarding the tech contribution, I feel the new addition for model and reward is small. The sub-sampling indeed address the large space problem empirically. The reason is that I have thought how to address this large space problem by myself. It is nice that the author try this simple way and it works. I will raise my score for this point.

---

> > > ### Author Response · Authors · 2024-11-25
> > > **Thank you for increasing score!**
> > >
> > > Dear reviewer,
> > >
> > > We thank the reviewer for increasing the score with positive response to our primary contribution, sub-sampling.
> > >
> > > Please let us know if there are any additional questions!

---

> > > ### Author Response · Authors · 2024-11-26
> > > **Official Comment by Author**
> > >
> > > Dear Reviewer ovVP,
> > >
> > > We would like to emphasize that we have updated Section 4.6 with the additional analysis on **3D interaction modeling** for future work, in additional response to the Weakness 1: "insufficient characterization of 3D interactions".
> > >
> > > In this experiment, we found that the model performance can be improved by simply incorporating the protein-ligand binding conformation information for each state molecule while maintaining the current graph transformer structure. Below are the Top 100 Vina scores according to the training steps. The binding conformations were predicted using the GPU-accelerated docking tool UniDock [1].
> > >
> > > | Training iterations | 100 | 200 | 300 | 400 | 500 | 600 | 700 | 800 | 900 | 1000 |
> > > | --- | --- | --- | --- | --- | --- | --- | --- | --- | --- |  --- |
> > > |w/o 3D interaction modeling |-3.96±0.80 | -10.99±0.21 | -11.48±0.29 | -11.81±0.30 | -12.11±0.27 | -12.24±0.24 | -12.35±0.19 | -12.44±0.16 | -12.50±0.14 | -12.55±0.13 |
> > > |w/ 3D interaction modeling |-5.38±0.58 | -10.96±0.34 | -11.51±0.29 | -12.01±0.20 | -12.26±0.12 | -12.44±0.05 | -12.52±0.06 | -12.61±0.05 | -12.68±0.05 | -12.71±0.06 |
> > >
> > > We believe this observation lays a groundwork for integration between synthesis-oriented generation and 3D-based structure-based molecular generation. For instance, this could involve using binding prediction methods, as demonstrated in this experiment [1,2,3], or directly generating 3D binding conformations from pocket structures like existing structure-based drug design (SBDD) methods.
> > >
> > > We hope this additional result addresses your concerns more clearly, and we look forward to your feedback during the extended discussion period.
> > >
> > > The authors
> > >
> > > ---
> > > **References:**
> > > 1. Yu, Yuejiang, et al. "Uni-Dock: GPU-accelerated docking enables ultralarge virtual screening." Journal of chemical theory and computation 19.11 (2023): 3336-3345.
> > > 2. Corso, Gabriele, et al. "Diffdock: Diffusion steps, twists, and turns for molecular docking." arXiv preprint arXiv:2210.01776 (2022).
> > > 3. Abramson, Josh, et al. "Accurate structure prediction of biomolecular interactions with AlphaFold 3." Nature (2024): 1-3.

---

> ### Author Response · Authors · 2024-11-22
> **Response to Reviewer ovVP (2/n)**
>
> Here is the concise summary of the comparison except for reward setting which is depending on optimization objectives:
> | Method                       | SynFlowNet(v1)                      | SynFlowNet(v2, after ICLR deadline)                      | Ours                                  |
> |------------------------------|---------------------------------------|---------------------------------------|---------------------------------------|
> | Model framework              | No action embedding, Hierarchical MDP | Action embedding, Hierarchical MDP    | Action embedding, Non-hierarchical MDP |
> | How to prevent invalid backward transitions | No                                | Additional training objectives        | Explicit retrosynthetic analysis      |
> | Sampling algorithm           | No additional technique for large action space | No additional technique for large action space | Action space subsampling            |
>
> ---
> ---
>
> **Question 1: For conventional SBDD methods, they are generally trained on molecules that can bind to protein pockets, resulting in generated small molecules with a specific distribution. I am curious about how the molecules generated in this space of synthesizable reaction-based small molecules differ from those generated by SBDD methods, such as pocket2mol. Because drug-likeness is not solely determined by synthesizability and vina score alone. For instance, the bond angle, number of rings. These facts are hard to encode into the reward function.**
>
> SBDD methods use MLE objectives to learn multiple properties inherent in the training dataset. However, it is difficult to represent all drug-like properties to an explicit reward function. **Therefore, our method restricted the sample space to the drug-like chemical space by assembling the drug-like building blocks.** It is well known that Enamine REAL [2], which is constructed from the Enamine building block library through the similar composing process as us, covers drug-like chemical space. If we replace the public reaction templates used in this work to the Enamine synthesis protocol, we can align the sample space to the Enamine REAL.
>
> For the first suggested metric, bond angle, our method does not construct 3D conformers. Therefore, it is difficult to compare with 3D generative models. Other important drug-like structural features are reported below:
>
> |            | Number of Aromatic Rings | Number of HBondDonor | Number of HBondAcceptor |
> |------------|--------------------------|----------------------|-------------------------|
> | Pocket2Mol | 1.598 ± 1.337            | 1.733 ± 1.305        | 3.766 ± 2.169           |
> | TargetDiff | 0.472 ± 0.705            | 3.266 ± 1.952        | 5.044 ± 2.601           |
> | DecompDiff | 1.401 ± 1.147            | 3.384 ± 2.003        | 6.906 ± 2.527           |
> | DiffSBDD   | 0.135 ± 0.388            | 2.231 ± 1.564        | 4.235 ± 2.183           |
> | MolCRAFT   | 1.486 ± 1.147            | 4.088 ± 2.460        | 6.257 ± 2.900           |
> | RxnFlow    | 1.489 ± 0.853            | 1.455 ± 0.961        | 4.711 ± 1.588           |
>
> **Question 2: Please compare the speed of different methods.**
>
> We thank the reviewer for suggesting additional comparative study.
> We included the speed benchmark (training, generation) according to the building block library size in Section D.3.
> The results demonstrate the cost-efficiency of our method compared to baseline GFlowNets.
>
> The tables below include the average of runtime per batch for 100 batch (SynFlowNet has limited up to 400,000 building blocks due to a memory limitation.):
> 1. Training Time
> |    Num Blocks     | 10,000 | 100,000 | 200,000 | 300,000 | 400,000 | 500,000 | 600,000 | 700,000 | 800,000 | 900,000 | 1,000,000 |
> |----------------|--------|---------|---------|---------|---------|---------|---------|---------|---------|---------|-----------|
> | SynFlowNet    | 0.72   | 1.16    | 1.72    | 2.21    | 2.73    | -       | -       | -       | -       | -       | -         |
> | RGFN          | 0.56   | 0.70    | 0.99    | 1.38    | 1.45    | 1.74    | 1.92    | 2.17    | 2.46    | 2.60    | 2.99      |
> | RxnFlow       | 1.12   | 1.14    | 1.17    | 1.20    | 1.18    | 1.18    | 1.20    | 1.22    | 1.23    | 1.22    | 1.22      |
> 2. Sampling Time
> | Num Blocks          | 10,000 | 100,000 | 200,000 | 300,000 | 400,000 | 500,000 | 600,000 | 700,000 | 800,000 | 900,000 | 1,000,000 |
> |----------------|--------|---------|---------|---------|---------|---------|---------|---------|---------|---------|-----------|
> | SynFlowNet    | 0.98   | 1.11    | 1.29    | 1.44    | 1.62    | -       | -       | -       | -       | -       | -         |
> | RGFN          | 0.68   | 0.84    | 1.11    | 1.25    | 1.47    | 1.64    | 1.90    | 2.09    | 2.62    | 2.37    | 2.74      |
> | RxnFlow       | 0.93   | 0.95    | 0.97    | 0.99    | 0.97    | 1.01    | 1.04    | 1.11    | 1.06    | 1.08    | 1.13      |

---

> ### Author Response · Authors · 2024-11-22
> **Response to Reviewer ovVP (3/n)**
>
> **Question 3: Why use 3 reaction steps for your method and 4 steps for SynFlowNet and RGFN?**
>
> We followed the settings from the original paper on RGFN and SynFlowNet. Since they use restricted building block (BB) sets (RGFN: 0.03%, SynFlowNet: 0.5% of an entire BB library), their sample spaces are extremely constrained compared to ours. For example, RGFN can access $2\times10^8$ molecules with 3 reaction steps while ours can access $7\times10^{11}$ molecules with only 1 reaction step. To extend the search space with a restricted BB set, they considered more than 3 reaction steps in their paper.
>
> ---
> **Reference:**
> 1. Jain, Moksh, et al. "Multi-objective gflownets." International conference on machine learning. PMLR, 2023.
> 2. Grygorenko, Oleksandr O., et al. "Generating multibillion chemical space of readily accessible screening compounds." Iscience 23.11 (2020).

---

### Official Review · Reviewer_sR3E · 2024-10-31

**Soundness:** 3
**Presentation:** 3
**Contribution:** 3
**Rating:** 6
**Confidence:** 4

**Summary:**

This paper proposes  a GFlowNets based method 'RXNFLOW' for drug discovery. By employing space subsampling technique, RXNFLOW can expand the search space and handle massive action spaces. This method can be adopted for pocket-specific optimization task as well as pocket-conditional generation task.

**Strengths:**

1. The method enables the generation of synthetic pathways for molecules, allowing for the sampling of highly synthesizable compounds while maintaining a significant level of diversity, which is meaningful for drug discovery.
2. With the enhancement of building blocks, the method demonstrates good scalability.
3. The experiments were conducted thoroughly, and the presentation is relatively clear.

**Weaknesses:**

1. Regarding the pocket-conditional generation task, to my knowledge, the more advanced methods [1] have not been compared.  This has somewhat affected the persuasive power of the experiments.
2. Compared to other SBDD methods, it seems that direct generation of conformations combined with the pocket is not achievable.
3. Regarding the pocket-specific optimization task, I notice that the reward function consits of Vina Score which is also used for evaluation. I have concerns about whether the method may be overfitting to Vina, which could result in inflated evaluation metrics.


[1]. Qu, Yanru, et al. "MolCRAFT: Structure-Based Drug Design in Continuous Parameter Space." ICML 2024.

**Questions:**

1. What limitations do you perceive in the current method, and how might it be improved in the future?
2. Regarding the pocket-conditional generation task, I am curious about the method's ability to synthesize molecules with specific binding characteristics, such as calculating the 'delta score' metric propsed in Paper [1].


[1]. Gao, Bowen, et al. "Rethinking Specificity in SBDD: Leveraging Delta Score and Energy-Guided Diffusion." ICML 2024.

**Details Of Ethics Concerns:**

NA.

---

> ### Author Response · Authors · 2024-11-22
> **Response to Reviewer sR3E (1/n)**
>
> We appreciate the reviewer for valuable feedbacks. We included the response for each question sequentially.
>
> **Issue 1: "Regarding the pocket-conditional generation task, to my knowledge, the more advanced methods have not been compared. This has somewhat affected the persuasive power of the experiments."**
>
> We thank the reviewer for suggesting the new state-of-the-art method for comparative study. In the revised manuscript, we include DiffSBDD and MolCRAFT (bayesian flow network) in Table 5.
>
> | Model        | Valid (↑) | Vina Avg. (↓) | Vina Med. (↓)  | QED Avg. (↑) | QED Med. (↑) | Synth Avg. (↑) | Synth Med. (↑) | Div Avg. (↑) | Time  Avg. (↓) |
> |--------------|-----------|----------------|----------------|--------------|--------------|---------------------|---------------------|--------------|---------------|
> | Pocket2Mol   | 98.3%     | -7.60         | -7.56          | 0.57         | 0.58         | 29.1%              | 22.0%              | 0.87         |    2504s    |
> | TargetDiff   | 91.5%     | -7.37         | -7.56          | 0.49         | 0.49         | 9.9%               | 3.2%               | 0.84         |     3428s    |
> | DecompDiff   | 66.0%     | -8.35         | -8.35          | 0.37         | 0.35         | 0.0%               | 0.0%               | 0.84         | 6189s            |
> | DiffSBBD     | 76.0%     | -6.95         | -7.10          | 0.47         | 0.48         | 2.9%               | 2.0%               | 0.88         | 135s           |
> | MolCRAFT     | 70.8%     | -9.25         | -9.24          | 0.45         | 0.44         | 3.9%               | 0.0%               | 0.82         | 131s           |
> | TacoGFN      | 100.0%    | -8.24         | -8.44          | 0.67         | 0.67         | 1.3%               | 1.0%               | 0.67         | 4 s            |
> | RXNFlow      | 100.0%    | -8.85         | -9.03          | 0.67         | 0.67         | 34.8%              | 34.5%              | 0.81         | 4 s            |
>
>
> **Issue 2: "Compared to other SBDD methods, it seems that direct generation of conformations combined with the pocket is not achievable."**
>
> We appreciate the reviewer for this insightful comment. Such 3D SBDD methods and the optimization methods (Reinforcement Learning(RL), GFlowNet) are orthogonal. Conventional SBDD focused on modeling of 3D structural conformations in the generation process, where GFlowNet is a training method for probability inference by modeling the energy-based unnormalized density function under the fixed reward function. Usually SBDD is trained with Maximum Likelihood Estimation (MLE) to learn given data distribution, and GFlowNets are alternatives to the MLE objective, not SBDD architectures. Thus, the development of 3D SBDD methodologies and optimization frameworks target different components of generative modeling and are complementary.
>
> In this context, we would like to emphasize that the primary goal of this work is not exploring better methods for modeling the protein-ligand interaction, but rather proposing a cost-efficient learning technology for scaling GFlowNet to handle massive synthetic action spaces.
>
> **Issue 3: "Regarding the pocket-specific optimization task, I notice that the reward function consits of Vina Score which is also used for evaluation. I have concerns about whether the method may be overfitting to Vina, which could result in inflated evaluation metrics."**
>
> In Tables 1-4, we conducted optimization using the QED/Vina reward function for both our method and the baseline methods to ensure a fair comparison. Optimization algorithms such as Genetic Algorithm (GA), Reinforcement Learning (RL), and GFlowNet aim to discover high-reward samples under a given reward function, which is different to distribution learning-based methods to model a given data distribution.
>
> In molecular discovery, these optimization methods are developed for replacing the brute-force virtual screening, which identifies the top-scoring molecules using docking. Consequently, the baseline optimization studies (SynNet, GFlowNet, RGFN, SynFlowNet) and the molecular optimization benchmark [1] used the same reward function for both training and evaluation.

---

> ### Author Response · Authors · 2024-11-22
> **Response to Reviewer sR3E (2/n)**
>
> **Question 1: "What limitations do you perceive in the current method, and how might it be improved in the future?"**
>
> In the revised manuscript, we have included the two directions of possible future improvements in the conclusion section.
> 1. We can incorporate the 3D protein-ligand interaction modeling in this synthesis-oriented generation framework. Since binding affinity is closely related to the binding structure, integration with binding conformation generation methods can not only improve performance but also make the decision process more reasonable and interpretable [2]. One line to integrate these methods is using pre-trained binding conformation generation tools such as molecular docking or DL-based docking [3]. Other line is incorporating with 3D auto-regressive generative models such as Pocket2Mol. In both case, the mode-collapse problem of sequential generative methods should be addressed.
> 2. Our current action space subsampling uniformly samples the building block library to minimize bias. However, some building blocks are not suitable for the interest target protein, so uniformly sampling all building blocks may limit the optimization performance. Therefore, we can consider for introducing a prioritization method for building blocks instead of uniform sampling (introducing a bias) to enhance the exploitation.
>
> **Question 2. "Regarding the pocket-conditional generation task, I am curious about the method's ability to synthesize molecules with specific binding characteristics, such as calculating the 'delta score' metric"**
>
> We appreciate the reviewer for suggesting the additional metric to evaluate the pocket specificity. We includes the delta score in Section D.5. However, due to the limitations of computational resources, we were only able to perform DecompDiff, TacoGFN, and our model. We promise to measure delta score for other baseline models, and include them in the camera ready version.
>
> | Category  | Model       | Delta Score |
> |-----------|-------------|-------------|
> | Atom      | DecompDiff  | -1.29       |
> | Fragment  | TacoGFN     | -1.13       |
> | Reaction  | RxnFlow     | -1.13       |
>
> ---
>
> **Reference:**
> 1. Gao, Wenhao, et al. "Sample efficiency matters: a benchmark for practical molecular optimization." Advances in neural information processing systems 35 (2022): 21342-21357.
> 2. Chan, Lucian, et al. "A multilevel generative framework with hierarchical self-contrasting for bias control and transparency in structure-based ligand design." Nature Machine Intelligence 4.12 (2022): 1130-1142.
> 3. Corso, Gabriele, et al. "Diffdock: Diffusion steps, twists, and turns for molecular docking." arXiv preprint arXiv:2210.01776 (2022).

---

> > ### Comment · Reviewer_sR3E · 2024-11-23
> >
> > Thank you for your detailed response and supplementary experiments whcih address most of my concerns. I will maintain my score while increase the confidence.

---

> > > ### Author Response · Authors · 2024-11-25
> > > **Response to the Reviewer Comment**
> > >
> > > We appreciate the reviewer for the positive review, and we're glad to hear that our additional experiments addressed most of concerns.
> > >
> > > If additional questions are remaining, please let us know. We'll try our best to address the questions during the discussion period.

---

> ### Author Response · Authors · 2024-11-26
> **Official Comment by Author**
>
> Dear Reviewer sR3E,
>
> We just want to let you know that we've updated Section 4.6 with the additional analysis on **the effect of 3D interaction modeling**, which may be related to Question 1 and Weakness 2.
> We hope that these updates help to address your questions and concerns more clear.
>
> Below is the Top 100 Vina scores according to the training steps. The 3D spatial relationship between the molecule and protein was obtained from molecular docking [1]. From this result, we show that our methodology can be improved by integrating with 3D SBDD generative modeling or binding conformation prediction methods [2-3].
>
> | Training iterations | 100 | 200 | 300 | 400 | 500 | 600 | 700 | 800 | 900 | 1000 |
> | --- | --- | --- | --- | --- | --- | --- | --- | --- | --- |  --- |
> |w/o 3D interaction modeling |-3.96±0.80 | -10.99±0.21 | -11.48±0.29 | -11.81±0.30 | -12.11±0.27 | -12.24±0.24 | -12.35±0.19 | -12.44±0.16 | -12.50±0.14 | -12.55±0.13 |
> |w/ 3D interaction modeling |-5.38±0.58 | -10.96±0.34 | -11.51±0.29 | -12.01±0.20 | -12.26±0.12 | -12.44±0.05 | -12.52±0.06 | -12.61±0.05 | -12.68±0.05 | -12.71±0.06 |
>
>
> Please let us know if we can provide any further clarifications during the extended discussion period.
>
> The authors
>
> ---
> **References:**
> 1. Yu, Yuejiang, et al. "Uni-Dock: GPU-accelerated docking enables ultralarge virtual screening." Journal of chemical theory and computation 19.11 (2023): 3336-3345.
> 2. Corso, Gabriele, et al. "Diffdock: Diffusion steps, twists, and turns for molecular docking." arXiv preprint arXiv:2210.01776 (2022).
> 3. Abramson, Josh, et al. "Accurate structure prediction of biomolecular interactions with AlphaFold 3." Nature (2024): 1-3.

---

### Official Review · Reviewer_dTeg · 2024-11-08

**Soundness:** 3
**Presentation:** 3
**Contribution:** 2
**Rating:** 5
**Confidence:** 4

**Summary:**

This paper introduces a new gflownets-based method RxnFlow for conducting the synthesizability-aware generation. To this end, the authors proposed to redefine the action space with Reaction templates and building blocks. To make the training procedure flexible, the authors involve a subsampling method for training RxnFlow in huge spaces.  The experiments over the SBDD benchmark have demonstrated the effectiveness of the RxnFlow for generating molecules with both good binding affinity and synthesizability.

**Strengths:**

1. The motivation of the paper is clear. I believe that apart from the generative approaches that formulate the problem as a constraint generation/projection, the proposed methods focus on an alternative perspective, i,e,  explicitly limiting the action space of gflownets, which should also be explored.

2. The paper introduces a simple yet effective approach, which is referred to as subspace sampling. The method takes a very simple formulation while it enables the reduction of complexity and enables feasible training of the algorithm.

3. The empirical study of the proposed framework is extensive, with both pocket conditioned/ specific ligand generation being considered.

**Weaknesses:**

1. Though I appreciate the methods with simplicity and effectiveness, I believe that a more systematic investigation and overview of the proposed methods is needed.  Based on the bias/variance discussion in the appendix, does the proposed approach conduct a variance/efficiency tradeoff, i.e. large variance for high efficiency? This is a little counterintuitive for me, could the authors discuss this further. From the ablation in Fig. 6, with sufficient steps, a larger subspace shows better performance. Does the key benefit of the proposed method is accelerating the optimization?

2. Some important baselines are suggested to be included in Table 5, for example [1] for atom-level SBDD. And the paper mentioned the recent relevant work [2], so why not include it?

3. I would like to suggest the author include the SA score as another metric for evaluating the synthesizability. I am curious to see whether SA is aligned with the used metrics in the paper.

I would like to adjust my score based on the response.

**Questions:**

Refer to above

---

> ### Author Response · Authors · 2024-11-22
> **Response to Reviewer dTeg (1/1)**
>
> We appreciate the reviewer for valuable feedback. We will respond to each of the reviewer's issues individually.
>
> **Issue 1.1: "Though I appreciate the methods with simplicity and effectiveness, I believe that a more systematic investigation and overview of the proposed methods is needed. Based on the bias/variance discussion in the appendix, does the proposed approach conduct a variance/efficiency tradeoff, i.e. large variance for high efficiency?"**
>
> Our method makes a cost-variance trade-off or a memory-variance trade-off, which is similar to negative sampling of natural language processing [3]. Through this trade-off, our action space subsampling enables GFlowNets to handle larger action spaces with small computing power. In the revised manuscript, we have included **scaling laws for GFlowNet performance** with ours and the baseline models in Section D.3.
> - **Memory-variance trade-off:** Baseline GFlowNets require memory capacity proportional to the number of actions, making it infeasible for them to operate on million-scale building block libraries. In contrast, our method can theoretically scale to any library size by leveraging subsampling to control memory usage.
> - **Cost-variancen trade-off:** Similarly, baseline GFlowNets experience a runtime increase proportional to the size of the building block libraries. On the other hand, our method can maintain a constant computational cost by simply adjusting the subsampling ratio. This enables exploration of larger action spaces without additional computational overhead.
>
> **Issue 1.2: "From the ablation in Fig. 6, with sufficient steps, a larger subspace shows better performance. Does the key benefit of the proposed method is accelerating the optimization?"**
>
> In Figure 6 and Section D.3, we showed that our model performance can be improved by using larger action space. We note that x-axis means the building block library size, not the subsampling size. This result supports our primary assumption: using more building blocks is advantageous for discovering more potent and diverse molecules. While larger action spaces typically require significantly more computational resources in existing GFlowNet frameworks, our method effectively addresses this challenge by mentioned trade-offs.
>
> In summary, **the key benefit of our proposed method lies in introducing new scaling methods for GFlowNets**, improving the exploration performance by simply expanding the action space while maintaining the computing cost. This is particularly important for scaling GFlowNets to million-scale building block libraries, such as eMolecules, which contains over 9 million building blocks.
>
> **Issue 2: Suggestion on additional baselines.**
>
> We apologize for not fully addressing the reviewer's points in our first response. The references in the review ([1], [2]) seem to be missed, could I know these references? We promise to include these references, either in the discussion period or in the camera ready version.
>
> Current state, we included DiffSBDD [4], which is mentioned in the related work of our manuscript, and the recent SBDD work MolCRAFT [5].
>
> | Model        | Valid (↑) | Vina Avg. (↓) | Vina Med. (↓)  | QED Avg. (↑) | QED Med. (↑) | Synth Avg. (↑) | Synth Med. (↑) | Div Avg. (↑) | Time  Avg. (↓) |
> |--------------|-----------|----------------|----------------|--------------|--------------|---------------------|---------------------|--------------|---------------|
> | Pocket2Mol | 98.3% | -7.60 | -7.56 | 0.57 | 0.58 | 29.1% | 22.0% | 0.87 | 2504s |
> | TargetDiff | 91.5% | -7.37 | -7.56 | 0.49 | 0.49 | 9.9%  | 3.2%  | 0.84 | 3428s |
> | DecompDiff | 66.0% | -8.35 | -8.35 | 0.37 | 0.35 | 0.0%  | 0.0%  | 0.84 | 6189s |
> | DiffSBBD   | 76.0% | -6.95 | -7.10 | 0.47 | 0.48 | 2.9%  | 2.0%  | 0.88 | 135s  |
> | MolCRAFT   | 70.8% | -9.25 | -9.24 | 0.45 | 0.44 | 3.9%  | 0.0%  | 0.82 | 131s  |
> | TacoGFN    | 100.0%| -8.24 | -8.44 | 0.67 | 0.67 | 1.3%  | 1.0%  | 0.67 | 4s    |
> | RXNFlow    | 100.0%| -8.85 | -9.03 | 0.67 | 0.67 | 34.8% | 34.5% | 0.81 | 4s    |
>
>
> **Issue 3: "I would like to suggest the author include the SA score as another metric for evaluating the synthesizability. I am curious to see whether SA is aligned with the used metrics in the paper."**
>
> We thank the reviewer for suggesting an additional analysis. In Section D.2 of the revised manuscript, we included the violin plots of SA Scores for FragGFN, RGFN, SynFlowNet, and RxnFlow.
>
> ---
>
> **Reference:**
>
> [1], [2]: Reference included in the review.
>
> [3] Mikolov, Tomas, et al. "Distributed representations of words and phrases and their compositionality." Advances in neural information processing systems 26 (2013).
>
> [4] Schneuing, Arne, et al. "Structure-based drug design with equivariant diffusion models." arXiv preprint arXiv:2210.13695 (2022).
>
> [5] Qu, Yanru, et al. "MolCRAFT: Structure-Based Drug Design in Continuous Parameter Space." arXiv preprint arXiv:2404.12141 (2024).

---

> ### Comment · Reviewer_dTeg · 2024-11-25
> **Thanks for your Response**
>
> Sorry for my mistake of missing the reference. Exactly as you anticipated,  I am suggested to include DiffSBDD and MolCRAFT. I appreciate the efforts to include the extra results. There is a follow-up question regarding the results:
> In Table F5 of DiffSBDD, the validity is approximately 95% across different settings while in the revised table it is stated to be 76% in the updated Table;
> Besides, Figure 3 of Molcraft, shows the validity could approach over 90% and it is reported to be 70.8% in the above Table.
> Why is that? Could the author further investigate it and explain the reason?
> It is important to check the inconsistency which could cast doubt over the reliability of the experiments.

---

> > ### Author Response · Authors · 2024-12-02
> > **Official Comment by Authors**
> >
> > Dear Reviewer dTeg,
> >
> > We would like to gently remind you that the author-reviewer discussion period will conclude in less than 26 hours.
> > If you feel that your concerns have been fully addressed in our previous response, we kindly ask you to consider revising the score.
> > We look forward to receiving your feedback on our revision and are readily available to address any remaining questions or issues.
> >
> > Warm regards,
> >
> > The Authors

---

> ### Author Response · Authors · 2024-11-25
> **Response to Reviewer dTeg**
>
> We thank the reviewer for the response and the follow-up questions regarding the reliability of reported values.
>
> **DiffSBDD:**
> While the validity reported in Table F5 of the DiffSBDD is approximately 97.10%, some of the generated molecules contain disconnected fragments (Connectivity: 78.27%, as stated in the Table F5 of DiffSBDD).
> To ensure fairness with the evaluation metric used for other baselines, we defined a valid molecule as one that is both valid and complete. Therefore, we reported the validity value for DiffSBDD while accounting for connectivity.
> > Page 6, Section 5.1 of MolCRAFT: **Sample Efficiency.** In order to make a practical comparison among non-autoregressive methods, we report the average Time and **Generation Success, with the latter defined as the ratio of valid and complete molecules versus the intended number of samples.**
>
> To clarify, we have replaced the metric "Validity" to "Generation Success" in Page 8 of revised manuscript as follows:
> - Moreover, we report the **Generation Success (%)** which is the percentage of unique RDKit-readable molecules **without disconnected parts**, ...
>
> **MolCRAFT:**
> For the MolCRAFT results, we considered the samples generated by the *Ours-large* in Table 2 of the MolCRAFT, since its vina score is state-of-the-art.
> By evaluating the generated samples from its official GitHub, we found that the validity for *Ours-large* was about 70%, whereas *Ours-small* and *Ours* achieved validity above 95%.
> In an initial response, we did not recognize this difference in validity between these settings, so we only reported for *Ours-large*.
> We acknowledge this oversight and appreciate the reviewer highlighting this point.
>
> In the revised manuscript, we have included both *Ours* and *Ours-large* in Table 5:
>
> | Model        | Succ. (↑) | Vina Avg. (↓) | Vina Med. (↓)  | QED Avg. (↑) | QED Med. (↑) | Synth Avg. (↑) | Synth Med. (↑) | Div Avg. (↑) | Time  Avg. (↓) |
> |--------------|-----------|----------------|----------------|--------------|--------------|---------------------|---------------------|--------------|---------------|
> | DiffSBDD   | 76.0% | -6.95 | -7.10 | 0.47 | 0.48 | 2.9%  | 2.0%  | 0.88 | 135s  |
> | **MolCRAFT**   | **96.7%** | **-8.05** | **-8.04** | **0.50** | **0.50** | **16.6%**  | **9.1%**  | **0.84** | **141s**  |
> | MolCRAFT-large   | 70.8% | -9.25 | -9.24 | 0.45 | 0.44 | 3.9%  | 0.0%  | 0.82 | >141s  |
> | RXNFlow    | 100.0%| -8.85 | -9.03 | 0.67 | 0.67 | 34.8% | 34.5% | 0.81 | 4s    |
>
> **Once again, we thank the reviewer for pointing out this ambiguous part. We hope that our response addressed the question satisfactorily above, but if you have any additional questions, please let us know.**

---

### Official Review · Reviewer_91bL · 2024-11-10

**Soundness:** 2
**Presentation:** 3
**Contribution:** 2
**Rating:** 6
**Confidence:** 4

**Summary:**

This paper introduces RXNFLOW, a generative framework that integrates synthesizability considerations into molecular generation for drug design. Following previous work, RXNFLOW addresses synthesizability by generating molecules using molecular building blocks, chemical reaction templates and a GFlowNet model. RXNFLOW aims to efficiently learn to sample from a large and complex action space, employing an action space subsampling technique to handle the combinatorial explosion of possible synthetic pathways. The authors claim that RXNFLOW achieves strong performance in pocket-specific and pocket-conditional generation tasks, with high synthesizability and diversity scores compared to previous models.

**Strengths:**

- RxnFlow performs well across all tasks/targets and the comparison of a reaction-based GFlowNet to SBDD models is welcome.
- The subspace sampling technique substantially reduces memory and computational complexity.
- The use of non-hierarchical action base combining reactions and build blocks is novel and well motivated.
- Article is generally very well written and the quality of figures is high.

**Weaknesses:**

**Weakness/points to be worked on:**

- The main contribution of the works is the non-hierarchical and continuous action space. There are many theoretical benefits to this, but the benefit of their method is not concretely controlled with respect to the building block data used and compute budgets.
    - I would recommend Figure 6 be amended with scaling laws for the SynFlowNet [1], RGFN [2] and SyntheMol [3] methods.
- Another fundamental claim is that the model can generalise to ‘’unseen” building blocks. While in theory their method should allow this, it has not been justified with experimental evidence.
    - As far as I can tell, this is only benchmarked in Figure 5 which looks at QED. This is a highly uninformative metric given how well generative models do on QED. Benchmarking the seen and unseen molecules on one of the targets for instance would be a better example.
    - Furthermore, a random split of the building blocks are used, given the high degree of redundancy in the Enamine Building Block set it is not surprising that performance does not change as these sets are highly similar. How does the model do on highly novel building blocks?
- While technically impressive, I see no practical utility gained in the case on not retraining on new building block sets as (i) these libraries at not updated regularly and (ii) the costs of training these models are so small one would prefer to retrain a model anyways.
    - The y-axis scales in Figure 6 are extremely narrow for many metrics and does not represent a meaningful change in values for Vina and Structural Diversity. For Figure 6b there is only a meaningful reduction in scaffold diversity when number of available building blocks is 10^2, an extremely smaller number and other works do not use.
    - Furthermore, RGFN [2] introduces a similar method to scale to large build block spaces and a proper comparison is not conducted.
- Some aspects of the evaluation could be improved:
    - What is the with per run variance for the docking scores for molecules in Tables 1-4? Is there really a significant difference between the methods? I would recommend including per run box plots of Vina scores in the appendix.
    - There is next to no discussion on the limitations of their model and possible future work
    - L458: The authors do not include statistical tests or error bars/standard deviation values in Table 5 yet claim ”RXNFLow achieves significant improvements in drug-related properties” in the CrossDocked experiments.
    - No effort is made to disentangle the technical changes of the authors model v.s. building block library size when comparing to baselines.
- The article is generally very well written but parts are not clear, for instance:
    - The fact that the continuous embedding space is based on chemical fingerprints is not mentioned in the main text.
    - Table 6:  No legend/keys are provided to indicate what the symbols mean.
- L429: “Since RXNFLOW explicitly considers synthesiz- ability, we exclude the SA score from the TacoGFN’s reward function as described in Sec. C.2.”.  I Could not find reference to SA score in Section C.2. Am I right in thinking the authors removed the SA reward from TacoGFN? In which case they ablated another model they are comparing to. Or have they removed it when training their model? Please clarify.
- No code is provided.

**Minor points:**
- L122: “...trained models with the GFlowNet objective”, is it not more accurate to say these works propose reaction-based GFlowNet models?
- I struggle to see the added value in Algorithm 1 in the main text. It does not aid in understanding the method. I recommend removing.
- Table 6 seems wrong?
- In Table 3, calling this metric “Percentage of Synthesizable Molecules” is misleading, as these are predicted not ground truth. I would prefer AIZynFinder success as done in the previous literature [1].
- I appreciate the use of the LIT-PCBA targets, but is there a reason the SEH Proxy used in the rest of the GFlowNet literature [1,2] was not used?
- Section B.1: The formulation sees to be highly inspired by SynFlowNet, if so please cite accordingly.




1. Cretu, Miruna, et al. "Synflownet: Towards molecule design with guaranteed synthesis pathways." arXiv preprint arXiv:2405.01155 (2024).
2. Koziarski, Michał, et al. "RGFN: Synthesizable Molecular Generation Using GFlowNets." arXiv preprint arXiv:2406.08506 (2024).
3. Swanson, Kyle, et al. "Generative AI for designing and validating easily synthesizable and structurally novel antibiotics." Nature Machine Intelligence 6.3 (2024): 338-353.

**Questions:**

- Table 5: Values in time column have ‘a’ and ‘b’ superscripts. What do these mean?
- Section 4.2: Can the authors clarify what is ‘zero-shot’ about this method?
- Do you use the same reward functions for methods when comparing to previous GFlowNets?

---

> ### Author Response · Authors · 2024-11-22
> **Response to Reviewer 91bL (1/2)**
>
> We appreciate the reviewer for the detailed and insightful reviews.
>
> **Issue 1: "I would recommend Figure 6 be amended with scaling laws for the SynFlowNet, RGFN and SyntheMol methods."**
>
> We thanks the reviewer for this valuable feedback. In this work, we explore how to scale GFlowNets using cost- and memory-efficient sampling techniques. To address this, we have included the scaling laws for GFlowNets (SynFlowNet (v1, workshop version), RGFN, and our method) in Section D.3. Since SynFlowNet and RGFN exhibit trends similar to ours, their memory consumption is proportional to the size of the building block sets. Within the same memory capacity, we found that SynFlowNet and RGFN are limited to scaling up to 100k and 500k building blocks, respectively. In contrast, our method can handle a 1M building block library by leveraging a memory-variance trade-off.
>
> **Issue 2: "Benchmarking the seen and unseen molecules on one of the targets for instance would be a better example. Furthermore, a random split of the building blocks are used, given the high degree of redundancy in the Enamine Building Block set it is not surprising that performance does not change as these sets are highly similar. How does the model do on highly novel building blocks?"**
>
> Thank you for suggesting the new meaningful experiments. We included the results on structurally different seen/unseen building blocks (BBs) using the QED-Vina multi-objective optimization in Section D.7. The unseen BBs have less than 0.5 Tanimoto similarity to all seen BBs. We found that the reward distributions of molecules generated from these two sets of BBs were similar.
>
> **Issue 3: "While technically impressive, I see no practical utility gained in the case on not retraining on new building block sets as (i) these libraries at not updated regularly and (ii) the costs of training these models are so small one would prefer to retrain a model anyways."**
>
> Enamine added more than 65,000 building blocks (BBs) from June (1,309,385) to November (1,375,309) this year. Moreover, recent libraries (xREAL, Synple 4.0) expand their BB libraries by combining BB libraries from different venues such as Enamine and eMolecules.
>
> In addition, the training cost depends on the reward function. Various real-world drug discovery pipelines use conventional docking such as GOLD and GLIDE, which require minutes for a single batch with 64 molecules. Moreover, the training cost can be increased with accurate scoring tools which can be advanced with the availability of computationally intensive induced fit docking using AlphaFold3. Therefore, we believe that the consideration of updates on BB libraries provides practical utility.
>
> **Issue 3.1: "The y-axis scales in Figure 6 are extremely narrow for many metrics"**
>
> The purpose of Figure 6 is to emphasize that using more actions does not reduce the performance of our generative model. To discover novel molecules that are different from existing drug molecules, many recent works are focusing on expanding the search space [1, 2, 3, 4]. In this context, using more building blocks is the easiest way to expand the search space beyond the known chemical space. However, considering more actions increases the difficulty of deep generative model training, and in fact, RGFN authors reported performance degradation on larger BB libraries in their paper.
>
> **Issue 3.2: "RGFN introduces a similar method to scale to large build block spaces and a proper comparison is not conducted."**
>
> In Section D.3 of the revised manuscript, we included scaling laws with baseline models SynFlowNet and RGFN.
>
> **Issue 4: Suggestion for additional detailed evaluations****
>
> In the revised manuscript, we included all of suggestions:
> - Seciton D.2 for "per run variance for docking score"
> - Page 10 Section 4.6, Figure 7 for "future work"
>   - integration of 3D information modeling
>   - introducing the bias to action space subsampling to enhance exploitation
> - Section D.4 for "statistical information of Table 5"
> - Section D.3 (scaling laws) for "the disentangle of the technical changes between reaction-based GFlowNets"
>
> **Issue 5: Clarification of the ambiguous parts**
>
> We appreciate the detailed review. We clarified these ambiguous parts in the revised manuscript:
> - Section 3.3 for "continuous embedding space is based on chemical fingerprints is not mentioned"
> - Table 6 for "No legend/keys are provided"
>
> **Issue 6: The reward function for TacoGFN**
>
> We apologize for this missing part. We included both the reward function of TacoGFN and the reward function we used in Section C.3. TacoGFN is trained on three objectives, docking score, QED, and SA score, and we used the model weight from the original paper. In contrast, our model is trained on two objectives: docking score and QED:
> - TacoGFN: use the model weight in official github, which is trained on original reward function (Vina-QED-SA)
> - RxnFlow: train the model with Vina and QED reward, excluding SA.

---

> ### Author Response · Authors · 2024-11-22
> **Response to Reviewer 91bL (2/2)**
>
> **Issue 7: "No code is provided."**
>
> We apologize for the inconvenience of not mentioning the location of the source code in the original manuscript. The source code and all results for reproduction were submitted together, and they can be found in supplemental material (.zip). We also included the anonymous github in Abstract.
>
> ---
>
> **Minor Points 1, 2, 4, 6**
>
> We thank the reviewer for pointing out these ambiguous points. We have reflected these suggestions in the revised manuscripts.
>
> **Minor Point 3: Table 6 seems wrong?**
>
> The updated SynFlowNet structure was released on 10/16, after the ICLR submission deadline. Therefore, it was impossible for us to reflect the improved structure in the submitted manuscript. In the revised manuscript, we have reported both versions of SynFlowNet in Section B.5.
>
> **Minor Point 5: the reason the SEH Proxy was not used?**
>
> The sEH proxy was trained on the AutoDock Vina scores of 300k random samples from fragment-based GFlowNet, and the training data distribution can be different to the Enamine REAL molecules. This means that the proxy might limit exploration in a vast chemical space beyond the training set. In fact, RGFN authors reported as: *"Interestingly, we also observe structural differences between sEH proxy and sEH docking, possibly indicating poor approximation of docking scores by the proxy model."* Based on this observation, we decided to perform all experiments with actual docking only.
>
> ---
>
> **Question 1: 'a' and 'b' in Table 5**
>
> We thank the reviewer for pointing out the missing part. $a$ and $b$ mean the runtimes were measured on NVIDIA A100 and NVIDIA RTX 3090 respectively, while our model's runtime was measured on NVIDIA RTX A4000. Since our method’s speed was measured on the slower environment, we removed these superscripts due to page limit and included it on the section C.5 in the revised manuscript.
>
> **Question 2: What is 'zero-shot'**
>
> The *zero-shot* means that our model sampled the molecules for given target pockets, which are not used in the training. For this case, our flow network was trained on the CrossDocked2020 training pockets, and its performance is on unique 100 pockets which is not included in the training set.
>
> **Question 3: Do you use the same reward functions for methods when comparing to previous GFlowNets?**
>
> Yes. For fair comparison, we use the same reward function according to multi-objective GFlowNets (MOGFN; Jain et al [5]) for all GFlowNet baselines. We clarified this in the revised manuscript.
>
> ---
> **Reference**
>
> 1. Lyu, Jiankun, John J. Irwin, and Brian K. Shoichet. "Modeling the expansion of virtual screening libraries." Nature chemical biology 19.6 (2023): 712-718.
> 2. Sadybekov, Arman A., et al. "Synthon-based ligand discovery in virtual libraries of over 11 billion compounds." Nature 601.7893 (2022): 452-459.
> 3. Gentile, Francesco, et al. "Artificial intelligence–enabled virtual screening of ultra-large chemical libraries with deep docking." Nature Protocols 17.3 (2022): 672-697.
> 4. Gorgulla, Christoph, et al. "Virtualflow 2.0-the next generation drug discovery platform enabling adaptive screens of 69 billion molecules." bioRxiv (2023): 2023-04.
> 5. Jain, Moksh, et al. "Multi-objective gflownets." International conference on machine learning. PMLR, 2023.

---

> ### Author Response · Authors · 2024-11-25
> **Official Comment by Authors**
>
> Dear Reviewer 91bL,
>
> In response, we have provided additional comparative studies between baselines, an analysis of generalization ability, detailed statistical information about existing experiments, and a discussion of possible future work. This would be good improvement to our manuscript, and we sincerely thank you for your detailed review.
>
> As the author-reviewer discussion phase will conclude in less than 44 hours, we would like to ask if there is any additional information or clarification we can provide that might influence your assessment. We believe our responses have addressed most of the concerns and questions raised in your initial review. If we have properly addressed your concerns, we sincerely hope you will reconsider the assessment of our submission.
>
> Thanks again for your valuable feedback.
>
> Best, Authors

---

> ### Author Response · Authors · 2024-12-02
> **Official Comment by Authors**
>
> Dear Reviewer 91bL,
>
> We kindly remind you that there are less than 26 hours left in the extended author-reviewer discussion period.
>
> We have included the additional experiments, statistics, clarification, and public anonymous code repository in our revisions and believe we have thoroughly addressed your concerns.
> If our responses have resolved your concerns, we kindly ask you to reconsider your evaluation of our work accordingly.
> If you have any remaining issues or questions, please do not hesitate to let us know at your earliest convenience.
>
> Warm regards,
>
> The authors

---

> > ### Comment · Reviewer_91bL · 2024-12-03
> >
> > I thank the authors for their effort in the rebuttal and apologies for the considerable delay in getting back.
> >
> > I have read the rebuttal and revised manuscript and I think the work has been improved. I also thank the authors for providing the code repository. Therefore, I have raised my score.

---

### Author Response · Authors · 2024-11-22
**Revised manuscript**

Dear reviewers,

We extend our sincere appreciation to the reviewers for your invaluable insights and constructive feedback.
We tried to reflect all the reviews, and below is a brief description of the major updates.
Also, we included an anonymous GitHub link in the Abstract. The code and data for reproducibility are still available in Supplementary Materials.

**Thanks again, and please let us know if you have any additional questions.**

| Updates/Experiments | Section|
| --- | --- |
| Analysis of the affect of 3D interaction modeling  | Section 4.6 |
| Detailed comparison between baseline GFlowNets | Appendix B.5 |
| Property distribution of generated samples on LIT-PCBA benchmark | Appendix D.2 |
| Scaling laws (speed, optimization power) with baselines | Appendix D.3 |
| Statistical information for CrossDocked benchmark | Appendix D.4 |
| Target specificity of generated samples | Appendix D.5 |
| Ablation studies of Non-hierarchical MDP | Appendix D.6 |
| Investigation of the generalization ability to unseen building blocks | Appendix D.7 |

---

### Meta-Review · Area_Chair_mWbt · 2024-12-20

**Metareview:**

The paper proposes RxnFlow, a model based on generative flow networks that is shown to scale to large compound spaces.

The main contribution of the work is in scaling up GFlowNets to handle very realistic and comprehensive molecular spaces defined by 71 reaction templates and 1.2 million building blocks. Beyond its class of models, I understand that relatively few generative methods were shown to effectively scale to such space. RxnFlow is able to find (in its space) highly active molecules (e.g. comparable to MolCRAFT) that are significantly more synthetically accessible (as judged by open-source retrosynthesis software).

This feat is achieved using very well executes techniques such as adding action embeddings. Concurrently, SynFlowNet paper has been updated using related techniques.

Three reviewers voted for marginally accepting the paper, while two maintained their marginally negative opinion of the paper. Reviewers have generally engaged in the rebuttal, in certain cases increasing their score.

The strong empirical results and the overall soundness of the approach warrant acceptance of the work. The paper opens a practical avenue for building de novo generative models on large building block spaces, enriching the practical toolbox in the early stage drug discovery. Thank you for your submission and it is my pleasure to recommend accepting the paper.

**Additional Comments On Reviewer Discussion:**

During the rebuttal, the authors addressed concerns by providing detailed comparisons to recent baselines like MolCRAFT and SynFlowNet, refining evaluation metrics, and conducting additional experiments, including ablation studies and delta score analysis. These efforts clarified the scalability and utility of RxnFlow, reinforcing its position as a promising contribution to drug discovery pipelines.

---

### Decision · Program_Chairs · 2025-01-22

Accept (Poster)